# A generalized higher-order correlation analysis framework for multi-omics network inference

**Weixuan Liu**[ID][1]*, **Katherine A. Pratte**[2], **Peter J. Castaldi**[3], **Craig Hersh**[3], **Russell P. Bowler**[4], **Farnoush Banaei-Kashani**[5], **Katerina J. Kechris**[ID][1]

**1** Department of Biostatistics and Informatics, University of Colorado Anschutz Medical Campus, Aurora, Colorado, United States of America, **2** Department of Biostatistics, National Jewish Health, Denver, Colorado, United States of America, **3** Channing Division of Network Medicine, Department of Medicine, Brigham and Women's Hospital, Boston, Massachusetts, United States of America, **4** Division of Pulmonary Medicine, Department of Medicine, National Jewish Health, Denver, Colorado, United States of America, **5** Department of Computer Science and Engineering, College of Engineering, Design and Computing, University of Colorado Denver, Denver, Colorado, United States of America

* weixuan.liu@cuanschutz.edu

**Data availability statement:** The TCGA breast cancer data used in the real data experiment

## Abstract

Multiple -omics (genomics, proteomics, etc.) profiles are commonly generated to gain insight into a disease or physiological system. Constructing multi-omics networks with respect to the trait(s) of interest provides an opportunity to understand relationships between molecular features but integration is challenging due to multiple data sets with high dimensionality. One approach is to use canonical correlation to integrate one or two omics types and a single trait of interest. However, these types of methods may be limited due to (1) unaccounted higher-order correlations existing among features, (2) computational inefficiency when extending to more than two omics data when using a penalty term-based sparsity method, and (3) lack of flexibility for focusing on specific correlations (e.g., omics-to-phenotype correlation versus omics-to-omics correlations). In this work, we have developed a novel multi-omics network analysis pipeline called Sparse Generalized Tensor Canonical Correlation Analysis Network Inference (SGTCCA-Net) that can effectively overcome these limitations. We also introduce an implementation to improve the summarization of networks for downstream analyses. Simulation and real-data experiments demonstrate the effectiveness of our novel method for inferring omics networks and features of interest.

## Author summary

Multi-omics network inference is crucial for identifying disease-specific molecular interactions across various molecular profiles, which can help give insight into the biological processes related to disease etiology. Traditional multi-omics integration methods

section is available at: http://linkedomics.org/data_download/TCGA-BRCA/. Clinical data and SOMAScan data are available through COPDGene (https://www.ncbi.nlm.nih.gov/gap/, ID: phs000179.v6.p2). RNA-Seq data are available through dbGaP (https://www.ncbi.nlm.nih.gov/gap/, ID: phs000765.v3.p2). Metabolon data is available at Metabolomics Workbench (https://www.metabolomicsworkbench.org/ ID: PR000907). These datasets are high-level access control clinical study data, and thus are only available upon request at https://sharing.nih.gov/accessing-data/accessing-genomic-data/how-to-request-and-access-datasets-from-dbgap. The source code for SGTCCA-Net is available at https://github.com/liux4283/SparseGTCCANet.

**Funding:** The COPDGene study (ClinicalTrials.gov ID: NCT00608764) is supported by grants from the National Heart, Lung, and Blood Institute (U01HL089897 and U01HL089856), by National Institutes of Health contract 75N92023D00011, and by the COPD Foundation through contributions made to an Industry Advisory Committee that has included AstraZeneca, Bayer Pharmaceuticals, Boehringer-Ingelheim, Genentech, GlaxoSmithKline, Novartis, Pfizer, and Sunovion. This work was also supported by the National Heart, Lung, and Blood Institute (NHLBI) of the National Institutes of Health under award number R01 HL152735 (KK, WL) and TransOmics for Precision Medicine (TOPMed) fellowship (WL). The funders had no role in study design, data collection and analysis, decision to publish, or preparation of the manuscript.

**Competing interests:** The authors have declared that no competing interests exist.

focus mainly on pairwise interactions by only considering two molecular profiles at a time. This approach overlooks the complex, higher-order correlations often present in multi-omics data, especially when analyzing more than two types of -omics data and phenotypes. Higher-order correlation, by definition, refers to the simultaneous relationships among more than two types of -omics data and phenotype, providing a more complex and complete understanding of the interactions in biological systems. Our research introduces Sparse Generalized Tensor Canonical Correlation Network Analysis (SGTCCA-Net), a novel framework that effectively utilizes both higher-order and lower-order correlations for multi-omics network inference. SGTCCA-Net is adaptable for exploring diverse correlation structures within multi-omics data and is able to construct complex multi-omics networks in a two-dimensional space. This method offers a comprehensive view of molecular feature interactions with respect to complex diseases. Our simulation studies and real data experiments validate SGTCCA-Net as a potential tool for biomarker identification and uncovering biological mechanisms associated with targeted diseases.

## 1. Introduction

### 1.1. Multi-omics data integration

The integration of multiple datasets with the same set of subjects has been actively explored in the field of machine learning, referred to as multi-view machine learning [1]. This approach has a broad range of applications, including clustering subjects based on the consensus of different views [2] and performing image annotation [3]. Furthermore, multi-view machine learning has been used extensively in the biomedical domain [4,5]. Recent advances in biomedical technologies have enabled the generation of high-throughput data at the molecular level, such as the genome, transcriptome, proteome, and metabolome. Collectively, these datasets are referred to as multi-omics data [6]. Traditionally, each omics dataset has been analyzed separately [7,8], which can result in the loss of important information about the connections between omics datasets. Multi-omics integration is an alternative approach that identifies shared and complimentary information across different molecular profiles [9]. Integrating multi-omics data can help uncover biological mechanisms and interactions at the molecular level, and has been used for various purposes such as disease subtyping [10,11], variable selection [12,13], network analysis [14,15] and prediction of biomarkers [6,16,17].

### 1.2. Multi-view dimension reduction

Dimension reduction is one of the more common goals of high dimensional data analysis. Many methods are proposed to achieve such a goal by projecting the original data into the lower-dimensional embeddings, which includes and is not limited to Principal Component Analysis (PCA) [18], t-Distributed Stochastic Neighbor Embedding (t-SNE) [19], and Uniform Manifold Approximation and Projection (UMAP) [20]. However, these methods are primarily designed for single-view datasets and do not inherently account for the complexity of multi-view data, in which multi-omics data is a particular example. In recent years, many multi-view dimensional reduction techniques have been proposed to address this limitation and effectively handle data with multiple views or modalities. For example, non-negative matrix factorization is implemented to cluster multiple data sets [21], and multi-view co-reduction is developed to preserve the locality within the view and the consistency between

the views in the lower dimensional embedding of each view [22]. While these methods are proven to be effective for some multi-view tasks, they are not specifically designed to identify feature relationships and network inference across different views, which is the primary goal for multiomics integration that uncovers the interaction between molecules with respect to a phenotype. Another set of dimension reduction methods for multiple data sets are based on Canonical Correlation Analysis (CCA) [23], which achieve this goal by finding the linear combination (canonical weight) that maximizes the correlation between two sets of data. In the context of CCA, there is typically more than one solution. These solutions collectively form a canonical weight matrix that is used to project each dataset into a lower-dimensional shared space. This projection approach is common in many multi-view methods for dimension reduction. The concept of using more than one solution to extract latent factors is shared by various multi-view dimension reduction techniques, each with its own specific methodology, but the extraction of multiple canonical weight solutions is specific to CCA. Given two data sets $X_1 \in \mathbb{R}^{N \times d_1}$ and $X_2 \in \mathbb{R}^{N \times d_2}$, the optimization function is:

$$\max_{w_1, w_2} \frac{w_1^T X_1^T X_2 w_2}{\sqrt{w_1^T X_1^T X_1 w_1 w_2^T X_2^T X_2 w_2}},$$
(1)

where $w_1 \in \mathbb{R}^{d_1 \times 1}$ and $w_2 \in \mathbb{R}^{d_2 \times 1}$ are two canonical weight vectors for $X_1$ and $X_2$, respectively. However, this formulation only considers two sets of data and is not generalized to multiple data. A multiple canonical correlation method was developed to maximize the sum of pairwise canonical correlations [24]. Suppose that there are $j = 1, \dots, K$ views, then the problem can be formulated as follows:

$$\max_{w_1, \dots, w_k} \sum_{i<j}^{K} w_i^T X_i^T X_j w_j \quad s.t. w_j^T X_j^T X_j w_j = 1 \ \forall j.$$
(2)

A penalty term can be added to the formulation above to achieve sparsity, which is called Sparse Multiple Canonical Correlation Analysis (SmCCA) [24]. Specifically, SmCCA introduces L1 penalty terms to the objective function, resulting in the following optimization problem:

$$\max_{w_1, \dots, w_k} \sum_{i<j}^{K} w_i^T X_i^T X_j w_j - \sum_{j=1}^{K} \lambda_j ||w_j||_1 \quad s.t. w_j^T X_j^T X_j w_j = 1 \ \forall j.$$
(3)

Here, $\lambda_j$ are tuning parameters controlling the level of sparsity for each view. The L1 penalty ($||w_j||_1$) shrinks some elements of the weight vectors to be exactly zero to guarantee feature selection. This sparsity is particularly useful in high-dimensional settings, as it identifies a subset of features that are most relevant for the canonical correlations across multiple views. Therefore, SmCCA combines the benefits of multi-view analysis with feature selection, enhancing interpretability.

Network analysis for molecular profiles is often used with dimension reduction methods like SmCCA to identify and visualize connections between features [14,15] for the purpose of inferring underlying biological interactions. In addition to canonical correlation analysis, regression is another approach that is implemented for multi-omics network inference, particularly for gene regulatory networks [25,26]. However, most methods do not incorporate an outcome or phenotype in the form of a quantitative trait. Canonical correlation analysis-based methods can accomplish this goal by expanding CCA to incorporate a phenotype ($Y$). SmCCNet was proposed to partition omics features into different network modules while

considering the phenotype(s) of interest [27] and using a scaled version of SmCCA with the following optimization:

$$(w_1, w_2, ..., w_k) = \arg\max_{\tilde{w}_1, \tilde{w}_2, ..., \tilde{w}_K}$$

$$\left( \sum_{\substack{i<j \\ i,j=1,2,...,K}} a_{i,j} \tilde{w}_i^T X_i^T X_j \tilde{w}_j + \sum_{i=1}^{K} b_i \tilde{w}_i^T X_i^T Y \right),$$

$$\text{subject to } \|\tilde{w}_j\|^2 = 1, \quad P_j(\tilde{w}_j) \leq c_j, \quad j = 1, 2, ..., K,$$

(4)

where $a_{i,j}$ and $b_i$ are scaling factors that place greater importance on pairwise combinations of views. For example, one may want to increase $b_i$ to increase the influence on the phenotype to determine the canonical weights, and $P(\cdot)$ is the lasso penalty parameter for sparsity [28], but other types of penalties can also be used. The optimal penalty parameter is selected through k-fold cross-validation. After that, the canonical weights can be extracted based on the objective function above to construct an adjacency matrix. Finally, to construct a network, hierarchical clustering can be implemented afterward to extract multiple multi-omics network modules. This method has been applied in various contexts including identifying phenotype-specific miRNA-mRNA and proteomics-metabolomics correlations and networks [29]. However, the existing SmCCNet method can only be adapted to two molecular profiles plus one single phenotype. When extending it to 3 or more omics data types, it can become computationally expensive due to the cross-validation step to find the optimal penalty parameter for each omics data set.

Another multi-omics analysis and network inference method Data Integration Analysis for Biomarker discovery using Latent variable approaches for Omics studies (DIABLO) [30] has a similar formulation to the sparse multiple canonical correlation analysis problems (Eq 4). Although the formulations of SmCCNet and DIABLO are similar, they have some differences: (1) In DIABLO, there is no lasso penalty, but the user can choose how many features to include as nonsparse for each data view instead; (2) DIABLO focuses more on the prediction of phenotype and biomarker discovery, while SmCCNet focuses more on graph learning of the interaction between biomarkers with respect to phenotype; (3) the adjacency matrix of SmCCNet is aggregated through canonical weights obtained from subsampling the data multiple times for a more robust solution, while the adjacency matrix of DIABLO is the subset of correlation matrix with only molecular features selected by the canonical correlation analysis.

## 1.3. Tensor canonical correlation analysis

As reviewed, the dimension reduction methods based on canonical correlation analysis only consider the summation of all pairwise correlation, including methods like SmCCNet and DIABLO. However, for multi-omics data with more than two molecular profiles, the correlation structure can be higher-order rather than pairwise, where higher-order correlation is defined as the simultaneous correlation among 3 or more features. This type of higher-order correlation can be captured by pairwise correlation if and only if all pairwise correlations are strong. However, in multi-omics data, it may not be the case that higher-order correlations also have strong pairwise correlations.

A particular variant of the multiple canonical correlation analysis extends the pairwise relationship to a higher-order relationship by maximizing the tensor canonical correlation [31], which captures higher-order correlations among multiple data sets and projects them into a shared lower dimensional embedding. A further extension to this tensor canonical

correlation analysis (TCCA) is to combine deep learning with tensor canonical correlation analysis to maximize the higher-order correlation between views in the non-linear lower-dimensional space for dimensional reduction[32] to allow for non-linearity.

## 1.4. Contributions

As reviewed, SmCCA captures pairwise correlation, while TCCA captures higher-order correlation. However, TCCA only captures the highest order correlation, but not all types of lower-order correlation structures of interest. In addition, sparsity is not considered in TCCA, which prevents the model from focusing only on the most relevant information in the data. Furthermore, TCCA has a high memory and computation cost, which is inefficient for high-dimensional omics data. Therefore, in this work, we present a new method for identifying multi-omics phenotype specifics network that is based on TCCA but with significant extensions by including correlations of higher and lower order simultaneously (Fig 1) and by incorporating sparsity. This new method is called Sparse Generalized Tensor Canonical Correlation Analysis for Network Analysis (SGTCCA-Net). In particular, we avoid repeatedly applying TCCA to each correlation structure, but instead use a simultaneous algorithm that accounts for all the correlation components.

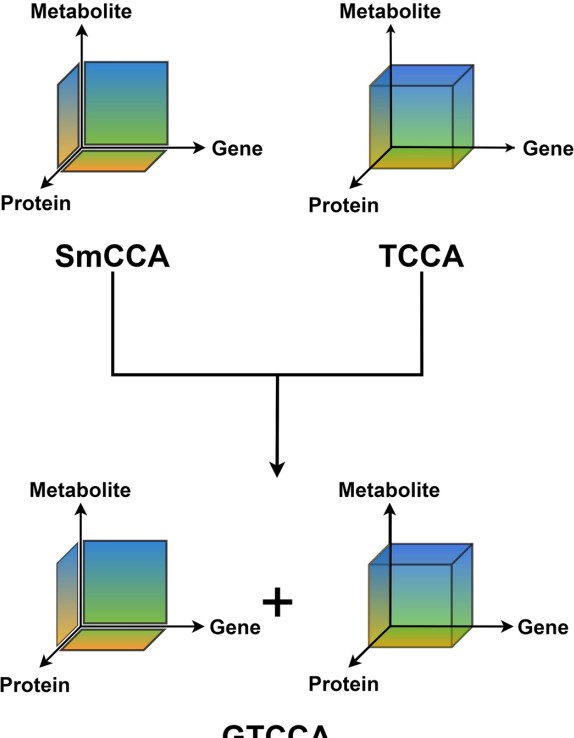

**Fig 1. Comparison between SmCCA, TCCA, and GTCCA.** Visualization of the comparison between Sparse multiple Canonical Correlation Analysis (SmCCA), Tensor Canonical Correlation Analysis (TCCA), and Generalized Tensor Canonical Correlation Analysis (GTCCA). SmCCA only captures pairwise correlations (e.g., genes and proteins) and TCCA only captures the highest order correlations (e.g., gene, protein and metabolites),. GTCCA considers the combination of highest order correlations ((TCCA) and lower-order correlations (SmCCA).

In this novel pipeline, we introduce multiple major contributions: New definition of higher-order correlation compared to the original TCCA, which better measures higher-order correlation without interpreting the directionality and avoids effect cancellation issues; The development of Sparse Generalized Tensor Canonical Correlation Analysis (SGTCCA) with consideration of higher-order correlation, flexible design of correlation structures, and biased subsampling algorithm that ensures sparsity and computational efficiency and implementation of network pruning and summarization algorithms to keep only the most important molecular features. Our pipeline first performs feature selection with SGTCCA, which extracts molecular features that are involved in one or more specified higher/lower-order correlation structures. Then an adjacency matrix between selected features is constructed to identify the inter-molecule relationships and collapse all higher/lower-order correlations into the two-dimensional space. Afterward, a network pruning algorithm is implemented to the adjacency matrix with the PageRank algorithm [33] and NetSHy summarization score [34] to reduce the size of the network and include only the most relevant molecular features, yielding the final multi-omics network. Details of the pipeline are described below

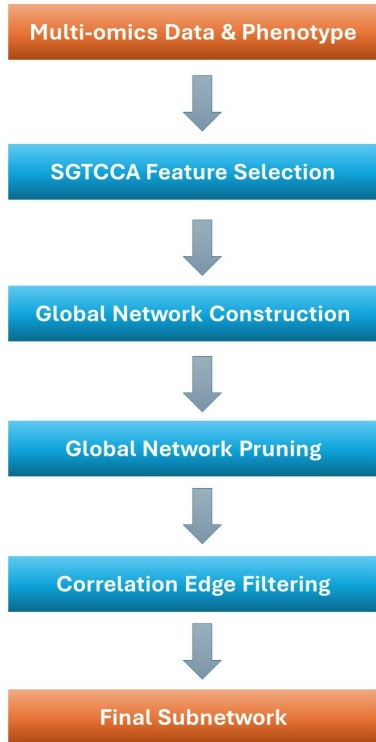

**Fig 2. SGTCCA-Net conceptual diagram.** Conceptual diagram of Sparse Generalized Tensor Canonical Correlation Network Analysis (SGTCCA-Net). It consists of four steps: (1) Sparse Generalized Tensor Canonical Correlation Analysis (SGTCCA) algorithm for feature selection; (2) Global networks construction based on results from (1); (3) Global network pruning; (4) Correlation edge filtering.

## 2. Materials and methods

### 2.1. Summary of pipeline workflow

Fig 2 illustrates the conceptual framework of the SGTCCA-Net algorithm. The process begins with the implementation of Sparse Generalized Tensor Canonical Correlation Analysis (SGTCCA) to perform feature selection, targeting higher and lower-order correlation structures of interest. The next step involves network analysis to identify interactions between molecular features across different molecular profiles, leading to the construction of global networks based on the SGTCCA results.

However, the global network may include molecular features that are less related to the phenotype of interest. To address this, a network pruning algorithm is applied to remove these irrelevant nodes. Following pruning, the resultant subnetwork may still contain weaker edges connecting less correlated molecular features. To refine the network further, an edge filtering

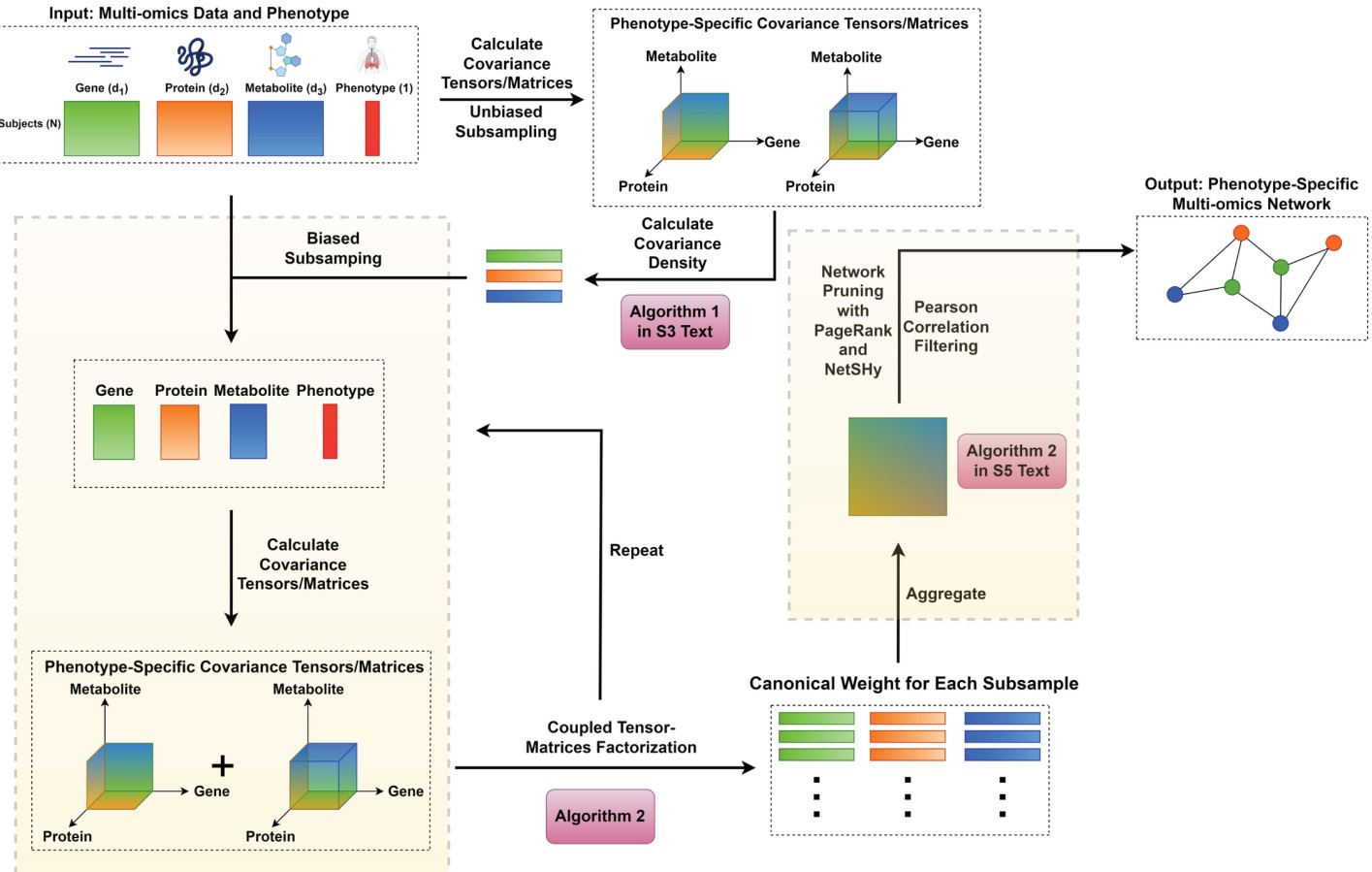

**Fig 3. SGTCCA-Net workflow.** Workflow of SGTCCA-Net pipeline for multi-omics network inference (made with BioRender: https://www.biorender.com). The workflow consists of four example input data: transcriptomics (gene), proteomics (protein), metabolomics (metabolite), and phenotype, each with $N$ subjects. Transcriptomics, proteomics, and metabolomics data have $d_1$, $d_2$, and $d_3$ features, respectively, and phenotype data has one feature. The pipeline first calculates the covariance density of each molecular feature, then the algorithm randomly selects a subset of the molecular features, biased towards features with high covariance density (Algorithm 1 in S3 Text, molecular features with high covariance density more likely to be selected). Based on the selected features, Generalized Tensor Canonical Correlation Analysis (GTCCA, Algorithm 2) is run to identify molecular features involved in higher/lower-order correlation of interest. Based on the GTCCA result, an affinity matrix between molecular features is constructed to identify the interaction between selected molecular features (Algorithm 2 in S5 Text). Network pruning then filters out weaker molecular features and edges in the affinity matrix, and network pruning (Algorithm 2 in S5 Text).

step based on Pearson's correlation is performed, eliminating weaker edges and enhancing the network's overall robustness.

The end-to-end pipeline for SGTCCA-Net (Fig 3 and Algorithm 1) inputs the multi-omics data $X_1, X_2, ..., X_k \in \mathbb{R}^{N \times d_j}, j = 1, 2, ..., k$, each with dimension $d_j$, and outputs subnetwork adjacency matrix $M_{(sub)} \in \mathbb{R}^{(\sum_{j=1}^{k} d_j^{(sub)}) \times (\sum_{j=1}^{k} d_j^{(sub)})}$, where $d_j^{(sub)} \leq d_j$ for all $j = 1, 2, ..., k$ is the number of features belong to the $j$th molecular profile. There are two major contributions in this work: (1) the development of Sparse Generalized Tensor Canonical Correlation Analysis that combines higher-order and lower-order correlation of interests and guarantees high computational efficiency, and (2) a novel network pruning algorithm based on the network summarization score.

Our pipeline is primarily designed for -omics data, such as transcriptomics, proteomics, metabolomics, and quantitative phenotypes associated with complex traits (see Fig 3). Additionally, it can handle other types of omics data, such as ATAC-Seq and epigenetics data. All omics datasets must include the same set of subjects ($N$) but can contain different numbers of molecular features, which may be either low or high-dimensional.

**Algorithm 1.**    SGTCCA-Net end-to-end pipeline algorithm.

---

**Input:** Multi-omics and phenotype data $X_j \in \mathbb{R}^{N \times d_j}, j = 1, 2, ..., k$, preferred number of subsamples $s$ ;
 (1) Run the SGTCCA algorithm and obtain canonical weight matrices $H_j \in \mathbb{R}^{d_j \times s}, j = 1, 2, ..., k$ **(Algorithm 2 and Algorithm 1 in Text C in S1 Appendix)**;
 (2) Construct global adjacency matrices $M$ based on $H_j, j = 1, 2, ..., k$ **(Algorithm 2 in Text E in S1 Appendix)**;
 (3) Prune each network module with PageRank algorithm and NetSHy network summarization score, and obtain sub-networks $M^{(sub)}$ **(Algorithm 2 in Text E in S1 Appendix)**;
 (4) Use Pearson's correlation to filter out weaker edges from each subnetwork  **(Algorithm 2 in Text E in S1 Appendix)**;
**Output:** Multi-omics network module with respect to phenotype(s)

---

## 2.2. Generalized tensor canonical correlation analysis

**2.2.1. New tensor canonical correlation analysis**  The original TCCA algorithm has two major issues: (1) while the magnitude of the higher-order correlation is informative, the sign (positive/negative) of the higher-order correlation is not interpretable for the direction of a relationship, especially when there are more than three datasets, and (2) the correlation effect will be canceled for the odd number of datasets (see Text A in S1 Appendix for detail). Therefore, we addressed these limitations by proposing a new tensor canonical correlation analysis formulation to (1) avoid the negativity of higher-order correlation value, and (2) calculate higher-order correlation value for the odd number of views with slight adjustments. While the directionality of higher-order correlation cannot be inferred directly, the directionality of interaction between any two features can be inferred during the network analysis step described later. Let $z_1, z_2, ..., z_k \in \mathbb{R}^{N \times 1}$ be $k$ $N \times 1$ dimensional vectors that are centered and scaled, and $\mathbf{1} \in \mathbb{R}^{N \times 1}$ is the all-one $N \times 1$ vector, then the higher-order covariance between these vectors can be defined as:

$$\rho(z_1, z_2, ..., z_k) = \begin{cases} |\frac{1}{n}(z_1 z_2 ... z_k)^T \mathbf{1}|, \\ \qquad\qquad \text{if } k = 2m, m \in \mathbb{N} \\ \frac{1}{n \cdot k} \sum_{j=1}^{k} |(z_1 ... |z_j| ... z_k)^T \mathbf{1}|, \\ \qquad\qquad \text{if } k = 2m + 1, m \in \mathbb{N}, \end{cases} \qquad (5)$$

where $z_1 z_2 ... z_k$ is the element-wise multiplication between $k$ vectors (assuming $k$ is even). The higher-order covariance calculation method differs slightly for the odd number of views to avoid effect cancellation (see Text A in S1 Appendix for detail). In this work, the higher-order covariance defined above is the unscaled measurement of higher-order correlation since it is difficult to mathematically scale to a 0-1 scale by extending Pearson's correlation definition. Therefore, throughout the paper, we use the higher-order covariance value to represent the higher-order correlation.

As mentioned above, the higher-order relationship may not imply multiple lower-order or pairwise relationships, and vice versa. Below, we provide some scenarios as an illustration. suppose there are 4 different feature vectors $z_1, z_2, z_3, z_4 \in \mathbb{R}^{N \times 1}$ that can be constructed by either (1) letting $z_1, z_2, z_3$ be less correlated with each other but strongly correlated with $z_4$ (e.g. this can be achieved when the correlation structure is $z_4 = z_1 + z_2 + z_3$), or (2) letting $z_1, z_2, z_3$ highly correlated with each other but independent of $z_4$. According to Eq 5, the 4-way higher-order covariance value between them can be regrouped to calculate the element-wise multiplication of the first three vectors, then element-wise multiplied by the last vector, which can be reformulated as:

$$\rho(z_1, z_2, z_3, z_4) = |\frac{1}{n}(z_1 z_2 z_3 z_4)^T \mathbf{1}| = |\frac{1}{n}(z_{1,2,3} z_4)^T \mathbf{1}|, \tag{6}$$

where $z_{1,2,3} = z_1 z_2 z_3$ is the element-wise multiplication of $z_1, z_2$ and $z_3$. In one particular scenario, a strong 4-way correlation between $z_1, z_2, z_3, z_4$ can be constructed by letting $z_1, z_2, z_3$ be independent of each other, and $z_4$ be $z_3$ plus a small random noise. In this case, the 4-way correlation between $z_1, z_2, z_3, z_4$ does not imply the 3-way correlation between $z_1, z_2, z_3$; In another particular scenario, a strong 3-way correlation between $z_1, z_2, z_3$ can be constructed by letting $z_1, z_2, z_3$ be strongly correlated with each other, but if $z_4$ is independent of $z_{1,2,3}$, then the 4-way correlation between $z_1, z_2, z_3, z_4$ does not exist.

Eq 5 calculates the covariance between multiple vectors, but if the covariance between matrices (e.g. each matrix is an omics dataset) needs to be calculated, we can avoid processing multiple loops to calculate the higher-order covariance for all combinations of features through a different equation. Suppose that there are $k$ data in total, denoted by $X_j \in \mathbb{R}^{N \times d_j}, j = 1, 2, .., k$, which are centered and scaled. Let $x_{ji} \in \mathbb{R}^{1 \times d_j}$ be the vector for subject $i$ in the $j$th view. For each view, there are $N$ observations and $d_j$ features, then the covariance tensor $C_{1,2,...,k} \in \mathbb{R}^{d_1 \times d_2 \times ... \times d_k}$ for $k$ views can be denoted by:

$$C_{1,2,...,k} = \frac{1}{N} |\sum_{i=1}^{N} x_{1i} \circ x_{2i} \circ ... \circ x_{ki}|, \tag{7}$$

where $\circ$ denotes the outer product. It can be shown that each entry of $C_{1,2,...,k}$, denoted by $C_{1,2,...,k}(j_1, j_2, ..., j_k)$, can also be calculated through Eq 5 (see Text A in S1 Appendix for more details). If the canonical weights for each view are $h_j \in \mathbb{R}^{d_j \times 1}, j = 1, 2, .., k$, then our new tensor canonical correlation analysis is formulated as follows:

$$\arg \max_{h_j} \rho = C_{1,2,...,k} \times_1 h_1^T \times_2 h_2^T \times ... \times_k h_k^T$$

$$\text{s.t. } h_j^T h_j = 1, j = 1, 2, ..., k, \tag{8}$$

where the $i$-mode product of $C_{1,2,...,k}$ and $h_i^T$, denoted by $C_{1,2,...,k} \times_i h_i^T$, is defined as follows:

$$(C_{1,2,...,k} \times_i h_i^T)(j_1, j_2, ..., j_{i-1}, j_{i+1}, ..., j_k) = \sum_{j_i=1}^{d_i} C_{1,2,...,k}(j_1, j_2, ..., j_i, ..., j_k) \cdot h_i(j_i), \quad (9)$$

where $h_j j_i$ is the $j_i$th entry of the vector $h_i$ and $C_{1,2,...,k} \times_i h_i^T \in \mathbb{R}^{d_1 \times ... \times d_{i-1}, \times d_{i+1} \times ... \times d_k}$. Therefore, when $k$ different $i$-mode product is performed ($i = 1, 2, ..., k$), $C_{1,2,...,k} \times_1 h_1^T \times_2 h_2^T \times ... \times_k h_k^T$ becomes a single number. It has been shown that the optimization problem above is equivalent to the following form [35]:

$$\min_{\rho, h_1, h_2, ..., h_k} \quad ||C_{1,2,...,k} - \hat{C}_{1,2,...,k}||_F^2$$
$$\text{s.t.} \quad \hat{C}_{1,2,...,k} = \rho \cdot h_1 \circ h_2 \circ ... \circ h_k. \quad (10)$$

In this optimization form, the canonical weights $h_j$ are solved using either a gradient-based method or alternating least squares [36].

### 2.2.2. Generalized tensor canonical correlation analysis combining higher and lower-order correlations

**Algorithm 2.**     Generalized tensor canonical correlation analysis algorithm.

```
Input: Multi-omics and phenotype data X_j ∈ ℝ^(N×d_j), j = 1, 2, ..., K, preferred number of
  network modules s ;
    (1) Calculate covariance tensors/matrices C_1, C_2, ..., C_l of interest using Eq
      7;
    (2) Formulate multiple tensors/matrices factorization problem using Eq 13
      ;
    (3) Use the first-order gradient-based algorithm to find optimal canonical
      weight matrices H_j ∈ ℝ^(d_j×s), j = 1, 2, ..., K;
Output: Canonical weight matrices H_j ∈ ℝ^(d_j×s), j = 1, 2, ..., k
```

The section above illustrates TCCA assuming that only a single full covariance tensor is used and is not capable of adding more covariance structures. For example, with the presence of transcriptomic, proteomics, metabolomics, and phenotype data, we may also be interested in the 3-way or pairwise correlation with respect to phenotype in addition to the full 4-way correlation structure. Therefore, to incorporate multiple correlation structures of interest, we extended our new TCCA method and developed the Generalized Tensor Canonical Correlation Analysis (GTCCA).

Let $S_m = \{(m_1, ..., m_m) : m_i \in \{1, ..., k\}, m_1 \neq m_2 \neq ... \neq m_m\}$ be all possible combinations of $k$ choose $m$, let $S_m(i)$ be the $i$th element in the set. For instance, with $k = 3$ and $m = 2$, it would be the set of $\{S_2(1) = (1, 2), S_2(2) = (1, 3), S_2(3) = (2, 3)\}$, then the GTCCA can be defined as:

$$\arg \max_{h_1, h_2, ..., h_k} a_{k,1} \rho_k(1)^2 + \sum_{j=1}^{\binom{k}{k-1}} a_{k-1,j} \rho_{k-1}(j)^2 + ...$$
$$+ \sum_{j=1}^{\binom{k}{3}} a_{3,j} \rho_3(j)^2 + \sum_{j=1}^{\binom{k}{2}} a_{2,j} \rho_2(j)^2$$
$$\text{s.t.} \quad h_j^T h_j = 1, j = 1, 2, ..., k, \quad (11)$$

where $\rho_m(j) = C_{S_m(i)} \times_1 h_{m_1} \times ... \times_m h_{m_m}$, where $(m_1, m_2, ..., m_m) = S_m(i)$. Compared to TCCA, this design allows flexibility with respect to a specific experimental design of interest by allowing a portion of $a_{i,j}$ to be 0. For example, if the investigator is not interested in a 3-way higher-order correlation between omics data 1,2, and 3, then the scaling factor associated with $\rho_{1,2,3}$ can be set to 0. In addition, the scaling factor $a_{i,j}$ can be set to values other than 1 to prioritize certain correlation structures. Eq 11 cannot be directly optimized with the existing gradient-based method, and thus we found the problem equivalency as follows (proof in Text B in S1 Appendix and is the direct extension of proof of tensor decomposition equivalency [35]):

**Theorem 1.** *Let $C_{S_m(j)}$ be the covariance tensor of view with $(m_1, ..., m_m) = S_m(j)$ such that $C_{S_m(j)} \in \mathbb{R}^{d_{m_1} \times d_{m_2} \times ... \times d_{m_m}}$, If the optimization goal is formulated as follows:*

$$\arg \max_{h_1, h_2, ..., h_k} a_{k,1} \rho_k(1)^2 + \sum_{j=1}^{\binom{k}{k-1}} a_{k-1,j} \rho_{k-1}(j)^2 + ...$$
$$+ \sum_{j=1}^{\binom{k}{3}} a_{3,j} \rho_3(j)^2 + \sum_{j=1}^{\binom{k}{2}} a_{2,j} \rho_2(j)^2$$
$$\text{s.t.} \quad h_j^T h_j = 1, j = 1, 2, ..., k, \tag{12}$$

*where $\rho_m(i) = C_{S_m(i)} \times_1 h_{m_1} \times ... \times_m h_{m_m}$ for all $m = 1, 2, ..., k$, then the optimization problem above is equivalent to the following:*

$$\arg \min_{h_1, h_2, ..., h_k} a_{k,1} ||C_{S_k(1)} - \hat{C}_{S_k(1)}||_F^2$$
$$+ \sum_{j=1}^{\binom{k}{k-1}} a_{k-1,j} ||C_{S_{k-1}(j)} - \hat{C}_{S_{k-1}(j)}||_F^2$$
$$+ ... + \sum_{j=1}^{\binom{k}{3}} a_{3,j} ||C_{S_3(j)} - \hat{C}_{S_3(j)}||_F^2$$
$$+ \sum_{j=1}^{\binom{k}{2}} a_{2,j} ||C_{S_2(j)} - \hat{C}_{S_2(j)}||_F^2$$
$$\text{s.t.} \quad h_j^T h_j = 1, j = 1, 2, ..., k, \tag{13}$$

*where $\hat{C}_{S_m(i)} = \rho_m(i) h_{m_1} \circ h_{m_2} \circ ... \circ h_{m_m}$ is the rank-1 approximation of $C_{S_m(i)}$.*

Since the original problem is shown to be equivalent to the multiple tensor factorization problem, it can be solved iteratively using any first-order gradient-based algorithm. We choose the non-linear conjugate gradient algorithm with the Dai/Yuan update and restart [37, 38].

## 2.3. Multi-omics data example

In this section, we demonstrate how the first-order gradient of Eq 13 can be calculated with a multi-omics data example. Suppose there are three types of molecular profiles: transcriptomics (tr), proteomics (pr), and metabolomics (me), and phenotype (ph) data. Define these

data as $X_{tr}, X_{pr}, X_{me}$ and $Y_{ph}$. Using GTCCA to find the phenotype-related correlation structure, the optimization problem is given by:

$$\arg \max_{h_{tr}, h_{pr}, h_{me}, h_{ph}} \rho^2_{tr,pr,me,ph} + \rho^2_{tr,pr,ph} + \rho^2_{tr,me,ph} + \rho^2_{pr,me,ph}$$
$$+ \rho^2_{tr,ph} + \rho^2_{pr,ph} + \rho^2_{me,ph}$$
$$\text{s.t. } h_j^T h_j = 1, j = tr, pr, m, ph, \tag{14}$$

where $\rho_{tr,pr,me,ph} = C_{tr,pr,me,ph} \times_1 h_{tr} \times_2 h_{pr} \times_3 h_{me} \times_4 h_{ph}$, and the other correlation components are also in the same form. By Theorem (1), optimizing this objective function is equivalent to:

$$\arg \min_{h_{tr}, h_{pr}, h_{me}, h_{ph}} ||C_{tr,pr,me,ph} - \hat{C}_{tr,pr,me,ph}||_F^2$$
$$+ ||C_{tr,pr,ph} - \hat{C}_{tr,pr,ph}||_F^2$$
$$+ ||C_{tr,me,ph} - \hat{C}_{tr,pr,ph}||_F^2$$
$$+ ||C_{pr,me,ph} - \hat{C}_{pr,me,ph}||_F^2$$
$$+ ||C_{tr,ph} - \hat{C}_{tr,ph}||_F^2 + ||C_{pr,ph} - \hat{C}_{pr,ph}||_F^2$$
$$+ ||C_{me,ph} - \hat{C}_{me,ph}||_F^2$$
$$\text{s.t. } h_j^T h_j = 1, j = tr, pr, m, ph, \tag{15}$$

where $\hat{C}_{tr,pr,me,ph} = \rho_{tr,pr,me,ph} \cdot h_{tr} \circ h_{pr} \circ h_{em} \circ h_{ph}$ represents the rank-1 approximation of the covariance tensor, and other components are similar. Below is the example gradient calculation for the transcriptome $h_{tr}$,

$$\frac{f}{h_{tr}} = \left[ \hat{C}_{tr,pr,me,ph(1)} - C_{tr,pr,me,ph(1)} \right]$$
$$\cdot \left( \rho_{tr,pr,me,ph} \cdot h_{tr} \odot h_{pr} \odot h_{me} \right)$$
$$+ \left[ \hat{C}_{tr,pr,ph(1)} - C_{tr,pr,ph(1)} \right] \left( \rho_{tr,pr,ph} \cdot h_{ph} \odot h_{pr} \right)$$
$$+ \left[ \hat{C}_{tr,me,ph(1)} - C_{tr,me,ph(1)} \right] \left( \rho_{tr,me,ph} \cdot h_{ph} \odot h_{me} \right)$$
$$+ \left( h_{tr} \rho_1 h_{ph}^T - C_{1,y} \right) h_{ph} \rho_{tr,ph} + \lambda \left( h_{tr} - \bar{h}_{ph} \right) \tag{16}$$
$$\frac{f}{\rho_{tr,pr,me,ph}} = \left( \hat{C}_{tr,pr,me,ph} - C_{tr,pr,me,ph} \right)$$
$$\times_1 h_{tr} \times_2 h_{pr} \times_3 h_{me} \times_4 h_{ph}, \tag{17}$$

where $\odot$ is the Khatri-Rao product, given two matrices $A \in \mathbb{R}^{m_1 \times n}$ and $B \in \mathbb{R}^{m_1 \times n}$, and let $\otimes$ denote the Kronecker product between two vectors, the Khatri-Rao product is given by:

$$A \odot B = \left[ a_1 \otimes b_1, a_2 \otimes b_2, ..., a_n \otimes b_n \right], \tag{18}$$

$C_{tr,pr,me,ph(i)}$ is the mode-$i$ matricization of tensor $C_{tr,pr,me,ph}$, and "mode" stands for the index of dimension in tensor data, and "mode-$i$" means the $i$th dimension of the tensor data. This is a way to matricize the tensor by mapping the elements of the tensor into a matrix by arranging the mode-i fibers (think of it as the higher-order rows and columns) so that they become the columns of $\left[ C_{tr,pr,me,ph} \right]_{(i)}$. The gradient for other $h$s and $\rho$s can be calculated in a similar way. After taking the gradient, the next step is to concatenate all of the gradients into a long vector. All first-order gradient-based methods can be used for optimization, and

the nonlinear conjugate gradient method is chosen to solve for the optimal canonical weights [39].

## 2.4. Sparse generalized tensor canonical correlation analysis

Common methods for ensuring sparsity in statistical models include using penalty-based techniques such as lasso [28] and elastic net [40]. However, applying these methods to GTCCA in practice can be computationally intensive, as they require tensor computation on the original covariance tensors, which can potentially cause memory issues. To address this problem, adapting the idea from Turbo-SMT [41], we use the biased subsampling method to guarantee sparsity (the details of the algorithm setup are shown in Algorithm 1 in Text C in S1 Appendix.). This method is both accurate and efficient, reducing the tensor factorization problem by 1000 times. The core concept of this method is to subsample features based on the covariance density value calculated for each feature. If a feature is more likely to be connected to other features or phenotypes, then it is more likely to be selected in the biased subsampling phase. Details of the calculation of the covariance density are given in Text D in S1 Appendix.

In practice, multi-omics data are often high-dimensional, which poses challenges in terms of memory and computation when computing covariance tensors. Our preliminary findings indicate that in GTCCA optimization, even a covariance tensor of size $1000 \times 1000 \times 1000$ requires more than 16 GB of RAM, leading to memory explosion. Although SGTCCA reduces the GTCCA memory requirement by feature subsampling, it still requires the calculation of covariance density vectors, which also involves the computation of the full covariance tensor. Therefore, to further improve the algorithm efficiency, when calculating the covariance density vectors, we used the unbiased feature subsampling approach to approximate each covariance tensor. In each subsampling iteration of our method, a small, randomly selected subset of features from each molecular profile is used. For these subsets, we calculate the covariance density based on the provided correlation structures. This process is repeated across multiple iterations. The final estimated covariance density vector is obtained by averaging the covariance densities from all subsamples. As a general rule for setting parameters, a lower subsampling percentage (e.g., 10%) requires a greater number of iterations to achieve reliable results, while a higher subsampling percentage (e.g., 70%) typically requires fewer iterations.

To ensure the consistency of canonical weight vectors from each subsample, a portion of features from each dataset are preselected and shared across different subsamples. For instance, for each subsample, we can have 8% of the features shared across all subsamples and 2% of the distinct features. A general recommendation is to set the total percentage of featured subsampled to be between 10% and 20% to ensure both computational efficiency.

## 2.5. Network construction and pruning

After obtaining the canonical weight through the SGTCCA, the next step is to construct an adjacency matrix. The adjacency matrix construction process is the same as SmCCNet, and the detailed algorithm is illustrated in Algorithm 2 of Text E in S1 Appendix. The general idea is to take the outer product between the concatenated canonical weight and itself. However, even though the adjacency matrix is sparse, it may still contain features/nodes that are less associated with other features/phenotypes. Therefore, we prune the global network with the PageRank algorithm [42] and the NetSHy network summarization score [34]. The original PageRank algorithm is widely used to rank web pages according to their importance. Its application to networks (adjacency matrix) is to count the number and strength of the edges for each node to determine the importance of each node. The NetSHy summarization score is the

principal component score that takes into account the graph/network topology. We chose the NetSHy summarization score rather than the regular principal component analysis to prioritize the contribution of nodes with high network connectivity. The network pruning algorithm ensures both a high correlation of the summarization score with the phenotype and high network connectivity. Step-by-step details of the algorithm can be found in Text E in S1 Appendix.

The algorithm above provides node-wise network pruning, but the pruned sub-networks will be highly densely connected, and it requires edge pruning. Therefore, we calculate the between-node Pearson's correlation and use it to filter out edges that connect to two weakly or non-correlated nodes, which improves the final network visualization.

## 2.6. Computational complexity

Suppose there are $k$ omics data, each with $d_1, d_2, ..., d_k$ features, the common subsampling proportion is $p_c$ and the distinct subsampling proportion is $p_d$ (see Text C in S1 Appendix for details). Define $p = p_c + p_d \in [0, 1]$, then the new tensor canonical correlation analysis defined in this paper has the space complexity of $O(d_1 d_2 ... d_k)$, while SGTCCA has the space complexity of $O(d_1 d_2 ... d_k p^k)$. The computational cost per iteration for new tensor canonical correlation analysis is $O(d_1 d_2 ... d_k s)$, while for SGTCCA it is $O(d_1 d_2 ... d_k p^k s)$. Since $0 < p < 1$, SGTCCA has a more efficient space complexity and computational cost than the new TCCA.

## 3. Experiments

### 3.1. Simulation study

We simulated multi-omics data using independent latent factors to represent various correlation structures [43] (Table 1). We aimed to compare the performance of SGTCCA-Net with

**Table 1. Simulated multi-omics data correlation structure for cases 1–3.** Red and "*" mean that features that are simulated with this latent factor are considered signal features. In addition to the existing latent factors and random noise, as shown in the table, additional random noise will be added to all simulated molecular features and phenotype data. The first table is for simulation case 1, where all types of phenotype-specific correlation structures are simulated and considered signal; the second table is for simulation case 2, where only 4-way phenotype-specific correlation structures are simulated and considered signal; the third table is for simulation case 3, where all 3-way, pairwise phenotype-specific correlation structure is simulated and considered signal.

| Data Type | Case 1: Feature Index | | | | | | | | | |
|---|---|---|---|---|---|---|---|---|---|---|
| | 1:20 | 21:40 | 41:60 | 61:80 | 81:100 | 101:160 | 161:220 | 221:280 | 281:340 | 341:1000 |
| Omics 1 | Latent 1* | Latent 2* | Latent 3* | Noise | Latent 5* | Latent 8 | Latent 9 | Latent 10 | Noise | Noise |
| Omics 2 | Latent 1* | Latent 2* | Noise | Latent 4* | Latent 6* | Latent 8 | Latent 9 | Noise | Latent 11 | Noise |
| Omics 3 | Latent 1* | Noise | Latent 3* | Latent 4* | Latent 7* | Latent 8 | Noise | Latent 10 | Latent 11 | Noise |
| **Phenotype Data Latent Factor Composition** | | | | | | | | | | |
| Phenotype | Latent 1*; Latent 2*; Latent 3*; Latent 4*; Latent 5*; Latent 6*; Latent 7* | | | | | | | | | |
| Data Type | Case 2: Feature Index | | | | | | | | | |
| | 1:20 | 21:40 | 41:60 | 61:80 | 81:100 | 101:160 | 161:220 | 221:280 | 281:340 | 341:1000 |
| Omics 1 | Latent 1* | Latent 2* | Latent 3* | Noise | Latent 5* | Latent 8 | Latent 9 | Latent 10 | Noise | Noise |
| Omics 2 | Latent 1* | Latent 2* | Noise | Latent 4* | Latent 6* | Latent 8 | Latent 9 | Noise | Latent 11 | Noise |
| Omics 3 | Latent 1* | Noise | Latent 3* | Latent 4* | Latent 7* | Latent 8 | Noise | Latent 10 | Latent 11 | Noise |
| **Phenotype Data Latent Factor Composition** | | | | | | | | | | |
| Phenotype | Latent 1* | | | | | | | | | |
| Data Type | Case 3: Feature Index | | | | | | | | | |
| | 1:20 | 21:40 | 41:60 | 61:80 | 81:100 | 101:160 | 161:220 | 221:280 | 281:340 | 341:1000 |
| Omics 1 | Latent 1* | Latent 2* | Latent 3* | Noise | Latent 5* | Latent 8 | Latent 9 | Latent 10 | Noise | Noise |
| Omics 2 | Latent 1* | Latent 2* | Noise | Latent 4* | Latent 6* | Latent 8 | Latent 9 | Noise | Latent 11 | Noise |
| Omics 3 | Latent 1* | Noise | Latent 3* | Latent 4* | Latent 7* | Latent 8 | Noise | Latent 10 | Latent 11 | Noise |
| **Phenotype Data Latent Factor Composition** | | | | | | | | | | |
| Phenotype | Latent 2*; Latent 3*; Latent 4*; Latent 5*; Latent 6*; Latent 7* | | | | | | | | | |

other multi-omics network inference pipelines. Since the comparison relies on the adjacency matrix, it is necessary that all methods have an adjacency matrix available. We generated three datasets for molecular profiles, termed omics 1, omics 2, and omics 3, along with a quantitative phenotype. To mimic the true multi-omics correlation structure in most scenarios (the strong connection between omics but the weak connection between omics and phenotype), we imposed weaker noise on the omics data and stronger noise on the simulated phenotype data, and all the random noise are generated from Gaussian distribution. This simulation study is designed to evaluate the accuracy of different methods in identifying omics features associated with both higher-order and lower-order phenotype-specific correlation structures, and to assess the robustness of the model in handling noisy or highly skewed omics data. To this end, we introduced three latent factors simulation strategies: multivariate normal latent factors, highly right-skewed latent factors (Fleishman power transformation [44]), and noisy latent factors (multivariate normal distribution with noise).

Under each latent factor simulation strategy, we proposed three distinct correlation structure setups (Case 1-3): Case 1 simulates all possible higher-order / lower-order phenotype-specific correlations (four-way, three-way, and pairwise correlations) as signal features; case 2 simulates only phenotype-specific four-way correlations as the signal features, and case 3 simulates all phenotype-specific three-way and pairwise correlation structures as the signal features. These signal correlation structures are simulated by latent factors mentioned above (red in Table 1). Furthermore, we also simulated non-phenotype-specific correlation structures to interfere with signal feature identification and test the robustness of each method (black in Table 1).

The positive "signal" features, intended for model identification, are defined as four-way correlations involving all omics and phenotype, multiple three-way correlations among different combinations of omics with phenotype, and pairwise correlations between each omics and phenotype. Additionally, non-phenotype-specific correlations are negative "random noise" features to challenge signal feature detection. These include a 3-way correlation among all omics and pairwise correlations among the omics, and these correlation structures serve as noise features, which models should ideally overlook. The performance is evaluated at the node level with Area Under the Receiver Operating Characteristic Curve (AUC) score. A node is predicted positive (signal) if its maximal connection to other nodes in the adjacency matrix passes a certain threshold, which is consistent with the SmCCNet evaluation [27]. The AUC score is then calculated checking prediction results across a series of the threshold value. Specifically, for each simulated feature, we identify and record the maximum edge value to other features in the adjacency matrix, treating this value as the predicted value. These predicted values are then scaled to fall within the 0-1 range. To evaluate the predictions, we compare them against the ground truth values, where each ground truth value is a binary indicator denoting whether a feature is a signal (1) or noise (0). The predictive performance is quantified using the Area Under the Receiver Operating Characteristic Curve (AUC-ROC). This involves sorting features by their predicted values, calculating the true positive rate (TPR) and the false positive rate (FPR) at various thresholds, plotting the ROC curve and computing the area under this curve. The AUC score effectively summarizes the model's ability to discriminate between signal and noise features, with a score of 1.0 indicating perfect discrimination and 0.5 indicating no discriminative ability (random guessing).

We compare the performance of SGTCCA-Net to other multi-omics network analysis methods with 25 replications: (1) STCCA-Net with only the higher order 4-way correlation structure;(2) Best SmCCNet AUC score from various combinations of scaling factors and sparsity levels; (3) Best DIABLO AUC score with combinations of different levels of sparsity and scaling factors. The parameter setup for SmCCNet and DIABLO resulted in a total of 9

models for each method. To ensure a fair comparison between methods, we extracted only one set of canonical weights from each method. More details on parameter settings can be found in Text F in S1 Appendix. SmCCNet and DIABLO are chosen because they are the only two methods that consider molecular interactions with respect to phenotype using a network approach. Both methods utilize a similar canonical correlation-based technique to generate the interaction network. By comparing SGTCCA-Net with these two methods, we can observe the direct improvements of SGTCCA-Net, as it incorporates higher-order correlations in the analysis.

## 3.2. Multi-omics data analysis

In addition to the simulations, we compared the performance of SGTCCA and SmCCA on two data sets. To maintain a fair comparison, we kept all subsequent network analysis steps constant, and therefore did not use DIABLO as the network inference for that method implements a different pipeline. Therefore, we opt for a modified version of SmCCNet (SmCCNet 2.0) [45], ensuring that the network analysis steps align with those of SGTCCA-Net. The best scaling factors and penalty terms selection for SmCCNet are based on 5-fold cross-validation. To be specific, we let the penalty term for each molecular profile vary between 0.25, 0.5, and 0.75, and the scaling factor associated with each phenotype-specific pairwise correlation component specific to the phenotype varies between 1, 5, and 10, and we evaluate the scaled prediction error for each particular combination. In an example of two omics data, if $(a_{1,2}, b_1, b_2)$ are the scaling factors where $b_1$ and $b_2$ represent the phenotype-specific scaling factors and $a_{1,2}$ stands for the scaling factor for between-omics correlation, we will scale the scaling factor by $\delta$ to ensure $a_{1,2} + b_1 + b_2 = 1$, then the best scaling factors and their associated penalty terms are determined by minimizing:

$$(a_{1,2}, b_1, b_2) = \arg \max_{\tilde{a}_{1,2}, \tilde{b}_1, \tilde{c}_2} \frac{|trainCC - testCC|}{|testCC|}, \tag{19}$$

where *trainCC* and *testCC* stand for total training canonical correlation and total testing canonical correlation. All other steps in the pipeline, including network construction and pruning, were kept unchanged. Specifically, we fix steps (2) through (4) of Algorithm 1 and vary the setup for step (1) between SGTCCA and SmCCNet. The study aimed to compare the performance of SGTCCA to SmCCA in extracting the network associated with the phenotype of interest.

**3.2.1. TCGA breast cancer network analysis**  The Cancer Genome Atlas Program (TCGA) breast invasive carcinoma project includes RNA sequencing data with normalized counts obtained through the Illumina HiSeq platform, microRNA (miRNA) expression data of tumor samples obtained through the Illumina HiSeq platform at the miRgene-level, and log-ratio normalized reverse phase protein arrays (RPPA) expression data from tumor samples at the gene level. In this experiment, we use tumor purity as the phenotype, defined as the percentage of cancer cells in a sample of tumor tissue [46]. After matching subjects with all available molecular profiles and phenotype data, we obtained a cohort of 105 subjects. Because of their relatively large number, we filtered genes based on the standard deviation to eliminate genes that exhibit lower variability and only include the top 25% of the genes, resulting in a total of 5039 genes, in addition to the 823 miRNAs, and 175 RPPAs. In the data preprocessing step, we regress out age, race, and whether patients received radiation therapy for each of the molecular features to adjust for covariates.

Our SGTCCA-Net pipeline assumes the correlation structure of gene-miRNA-RPPA-phenotype, gene-miRNA-phenotype, gene-RPPA-phenotype, miRNA-RPPA-phenotype, and all pairwise molecular profiles with phenotype. Same as the simulation study, we set the percentage of common subsampled molecular features at 8% and the percentage of distinct molecular features to 2% with a total of 10 iterations when bias subsampling is performed to guarantee sparsity. We use the network pruning algorithm to prune the multi-omics network and set the minimally/maximally final subnetwork size to 30/300.

To evaluate the efficacy of each method, we obtained the NetSHy summarization score for every subnetwork, and correlate the scores with tumor purity. In addition, we calculated the correlation of individual molecular features with tumor purity, contrasting the number of strongly and weakly correlated features between both methods. Since one of the advantages of SGTCCA-Net is its ability to extract higher-order correlation among molecular features, we rank all potential molecular feature triplets and pairs in relation to tumor purity based on their respective higher-order correlation values. This is done to investigate how different molecular features simultaneously interact with respect to tumor purity.

An enrichment analysis was also performed on the aggregate set of genes and proteins through Metascape (v. 3.5.20230501) [47], with the reference set of all genes and protein target genes fed into the SGTCCA-Net pipeline and recognized by Cytoscape (4540). While the aforementioned approach focused solely on genes and proteins, we also extended the enrichment analysis to miRNAs. MultiMiR is an R package and database with a collection of microRNAs/targets from external resources, including validated microRNA-target databases and predicted microRNA-target databases, which was used to evaluate microRNA-target gene associations in the inferred networks [48]. Using MultiMiR, we first identified validated target genes of miRNAs in the subnetwork, then treated target genes that have a non-zero canonical weight in the global network as the enriched set. Enrichment analysis is also performed in Metascape with the same background set as described above. Furthermore, to identify connections between enrichment results and cell type compositions, we also use cell type estimates from TCGA breast cancer data to find the association between the final subnetwork and cell type composition using CIBERSORT [49]. CIBERSORT is a computational method that utilizes gene expression data to estimate the abundances of specific cell types within a mixed cell population, providing information on the cellular composition of complex tissues [50]. Specifically, we take network PCs with significant associations to tumor purity and run Pearson's correlation between these PCs and the cell type estimates to identify which cell types are highly associated with the network.

**3.2.2. COPDGene network analysis** Phase II COPDGene [51] includes transcriptomics (RNA-Seq), proteomics (SomaLogic 1.3k), and metabolomics (Metabolon) data [52] for studying chronic obstructive pulmonary disease (COPD). For demonstration, we use a smaller cohort to compare different methods. We construct multi-omics networks with respect to forced expiratory volume at one second ($FEV_1$). To ensure that we have roughly 1000 genes to run the model, we use standard deviation to filter out genes with less variability (sd < 0.435). After filtering subjects with available data for all molecular profiles and the phenotype, there were 461 subjects with available data for all molecular profiles, with 972 genes, 1305 proteins, and 995 metabolites. To adjust for covariates, we regressed out effects from sex, age, and clinical center for each molecular feature.

Our SGTCCA-Net pipeline assumes the correlation structure of gene-protein-metabolite-phenotype, gene-protein-phenotype, gene-metabolite-phenotype, protein-metabolite-phenotype and all pairwise molecular profiles with phenotype. As in the simulation study and

the TCGA breast cancer study, we set the percentage of common subsampled molecular features at 8% and the percentage of distinct molecular features to 2% with a total of 10 iterations when biased subsampling was performed to guarantee sparsity.

To evaluate the efficacy of each method, we obtained the NetSHy summarization scores for every subnetwork, correlating them with $FEV_1$. In addition, we calculated the correlation of individual molecular features with $FEV_1$, contrasting the number of strongly and weakly correlated features between both methods. Since one of the advantages of SGTCCA-Net is its ability to extract higher-order correlation between molecular features, we rank all potential molecular feature triplets and pairs in relation to $FEV_1$ based on their respective higher-order correlation values. This is done to investigate how different molecular features simultaneously interact with respect to $FEV_1$.

Enrichment analysis was also performed on the union set of genes and proteins in the subnetwork through Metascape (v. 3.5.20230501) [47], with the background set to be all genes and protein target genes fed into the SGTCCA-Net pipeline and recognized by Cytoscape (1750). While the aforementioned approach focused solely on genes and proteins, we also extended the analysis to metabolites by running metabolite enrichment analysis on IMPaLa [53] based on the HMDB names of the metabolites in the subnetwork, with the background set to be all metabolites with available HMDB name (227).

$FEV_1$ is a measurement in liters for each subject, which did not take into account body habitus and race. Therefore, we also ran SGTCCA-Net on the $FEV_1$ percent predicted ($FEV_1PP$) to check if the top molecular features identified from $FEV_1$ are also identified in the $FEV_1PP$ network, and if the correlation changes. Additionally, we also ran an enrichment analysis in Metascape to check if the pathways overlapped between the two networks.

## 4. Results

### 4.1. Simulations demonstrate robustness and accuracy of SGTCCA-Net

SGTCCA-Net significantly outperforms the other three methods (SmCCNet, STCCA-Net, and DIABLO) in all simulation study settings and simulated correlation structure designs (Table 2), even when SmCCNet and DIABLO use the parameter setup that performs best in each iteration. In case 1, where all types of phenotype-specific correlation structures are presented and in case 3, when the 4-way correlation structure is removed, the performance of SGTCCA-Net improves by around 10% compared to the best-performing results from SmCC-Net and DIABLO, and the level of improvement increases as the number of subjects increases. SGTCCA-Net achieves high signal feature identification accuracy even with only 100 subjects in the presence and absence of different phenotype-specific correlation structures and provides nearly-perfect prediction when the number of subjects doubles, outperforming the other methods by around 15% to 20% in AUC. In case 2, when there is only one type of phenotype-specific correlation structure (4-way correlation), SGTCCA-Net, STCCA-Net, and best-performing DIABLO perform equally well with perfect or near-perfect AUC scores.

When the underlying latent factors are highly right-skewed and omics data are noisy, the performance of all models are impacted, but SGTCCA-Net still outperforms the other methods. This indicates that even when the signal is masked by noise to some extent, or normality is not guaranteed, SGTCCA-Net still performs well compared to other methods, and its performance improves as the sample size increases.

**Table 2. Simulation results.** Performance is evaluated through the AUC of the precision-recall curve generated by applying different thresholds to the maximal connection of molecular features to each other. For this simulation, 20 replications are the AUC median and interquartile range in parenthesis of is reported. "Best" AUC for SmCCNet and DIABLO denotes that in each replication, 9 SmCCNet/DIABLO models are run and only the highest AUC score is recorded. The first table is the simulation study for setting 1, which uses latent factors simulated from multivariate normal distribution; the second panel is the simulation study for setting 2, where latent factors are simulated with a highly right-skewed distribution; the third panel is the simulation study for setting 3, where latent factors are simulated with a multivariate normal distribution (same as case 1), but strong random noise is enforced on omics data. Each simulation study contains 3 cases: Case 1 means that the signal molecular features are defined as all 4-way, 3-way, and pairwise phenotype-specific correlation structure; case 2 removes the phenotype-specific 4-way correlation structure; case 3 removes the phenotype-specific 3-way and pairwise correlation structure. In each case, the data is simulated with 100 or 200 subjects.

| Normal Latent Factor: Median AUC (Interquartile Range) | | | | | |
|---|---|---|---|---|---|
| **Method** | **n** | **SGTCCA-Net** | **STCCA-Net** | **SmCCNet (Best)** | **DIABLO (Best)** |
| **Case 1** | **n = 100** | 0.885 (0.859, 0.906) | 0.656 (0.604, 0.680) | 0.706 (0.689, 0.736) | 0.745 (0.718, 0.780) |
| | **n = 200** | 0.978 (0.966, 0.986) | 0.694 (0.614, 0.738) | 0.742 (0.720, 0.818) | 0.825 (0.754, 0.880) |
| **Case 2** | **n = 100** | 1.000 (1.000, 1.000) | 0.995 (0.955, 1.000) | 0.825 (0.779, 0.853) | 0.931 (0.920, 0.954) |
| | **n = 200** | 1.000 (1.000, 1.000) | 1.000 (1.000, 1.000) | 0.823 (0.811, 0.837) | 0.939 (0.931, 1.000) |
| **Case 3** | **n = 100** | 0.890 (0.864, 0.942) | 0.622 (0.583, 0.701) | 0.710 (0.679, 0.740) | 0.701 (0.670, 0.757) |
| | **n = 200** | 0.977 (0.965, 0.983) | 0.679 (0.635, 0.711) | 0.766 (0.726, 0.816) | 0.814 (0.786, 0.891) |
| Skewed Latent Factor: Median AUC (Interquartile Range) | | | | | |
| **Method** | **n** | **SGTCCA-Net** | **STCCA-Net** | **SmCCNet (Best)** | **DIABLO (Best)** |
| **Case 1** | **n = 100** | 0.845 (0.814, 0.884) | 0.686 (0.603, 0.730) | 0.705 (0.664, 0.722) | 0.713 (0.675, 0.768) |
| | **n = 200** | 0.935 (0.921, 0.954) | 0.729 (0.675, 0.788) | 0.756 (0.726, 0.804) | 0.847 (0.813, 0.887) |
| **Case 2** | **n = 100** | 1.000 (1.000, 1.000) | 1.000 (0.980, 1.000) | 0.824 (0.780, 0.846) | 0.939 (0.932, 1.000) |
| | **n = 200** | 1.000 (1.000, 1.000) | 1.000 (1.000, 1.000) | 0.817 (0.770, 0.855) | 0.949 (0.933, 1.000) |
| **Case 3** | **n = 100** | 0.855 (0.799, 0.891) | 0.677 (0.614, 0.719) | 0.714 (0.681, 0.748) | 0.725 (0.651, 0.777) |
| | **n = 200** | 0.936 (0.925, 0.949) | 0.755 (0.678, 0.795) | 0.712 (0.687, 0.735) | 0.803 (0.738, 0.839) |
| Noisy Omics Data: Median AUC (Interquartile Range) | | | | | |
| **Method** | **n** | **SGTCCA-Net** | **STCCA-Net** | **SmCCNet (Best)** | **DIABLO (Best)** |
| **Case 1** | **n = 100** | 0.776 (0.761, 0.805) | 0.578 (0.554, 0.595) | 0.702 (0.677, 0.724) | 0.703 (0.667, 0.780) |
| | **n = 200** | 0.909 (0.892, 0.919) | 0.603 (0.579, 0.638) | 0.785 (0.748, 0.801) | 0.813 (0.766, 0.920) |
| **Case 2** | **n = 100** | 0.998 (0.996, 1.000) | 0.777 (0.666, 0.880) | 0.839 (0.789, 0.853) | 1.000 (0.918, 1.000) |
| | **n = 200** | 1.000 (1.000, 1.000) | 0.921 (0.822, 0.959) | 0.832 (0.815, 0.846) | 1.000 (0.936, 1.000) |
| **Case 3** | **n = 100** | 0.781 (0.746, 0.809) | 0.556 (0.538, 0.614) | 0.710 (0.678, 0.729) | 0.701 (0.670, 0.757) |
| | **n = 200** | 0.916 (0.888, 0.936) | 0.583 (0.565, 0.609) | 0.782 (0.745, 0.798) | 0.799 (0.778, 0.876) |

## 4.2. Relevant biological pathways found for TCGA breast cancer network analysis using SGTCCA-Net

After running the network analysis pipeline with SGTCCA and SmCCNet for the phenotype of tumor purity, the final network size for both SGTCCA-Net and SmCCNet is 300. Specifically, there are 128 genes, 125 miRNAs, and 47 proteins in the SGTCCA network, while there are 128 genes, 59 miRNAs, and 113 proteins in the SmCCNet network. The NetSHy network summarization score correlation with respect to tumor purity is 0.689 and 0.500 for SGTCCA-Net and SmCCNet respectively. These are relatively large values as the maximum and 90th quantile of the univariate correlation between any of the molecular features and tumor purity are 0.76 and 0.45 respectively. In the SGTCCA-Net network, there are 55 molecular features with weak correlation with respect to tumor purity (FDR-adjusted > 0.05), while for SmCCNet there are 105 such molecular features. To check whether each method includes these nodes incorrectly or whether they contribute to the network, we ran the NetSHy network summarization score on these weak nodes only, extracted the first 10 PCs, and calculated the maximum correlation between the PC score and tumor purity. For SGTCCA-Net, the maximum correlation is -0.232 ($p = 0.017$), while for SmCCNet, the maximum correlation is only -0.176 ($p = 0.072$). This implies that weak nodes within the SGTCCA-Net network

**Table 3. Top 5 molecular features from each molecular profile and their individual correlation with respect to tumor purity for TCGA breast cancer data (with p-value).**

| Type | Name | Correlation | FDR |
|---|---|---|---|
| Gene (128) | TMC8 | –0.712 | <0.001 |
| | ARHGAP9 | –0.752 | <0.001 |
| | SPIB | –0.638 | <0.001 |
| | HCST | –0.712 | <0.001 |
| | IRF8 | –0.748 | <0.001 |
| miRNA (125) | hsa-mir-142 | –0.564 | <0.001 |
| | hsa-mir-146a | –0.560 | <0.001 |
| | hsa-mir-150 | –0.593 | <0.001 |
| | hsa-mir-155 | –0.534 | <0.001 |
| | hsa-let-7i | –0.572 | <0.001 |
| Protein (47) | PCNA | –0.564 | <0.001 |
| | CCNB1 | –0.409 | <0.001 |
| | NRG1 | –0.466 | <0.001 |
| | KAT2A | 0.324 | 0.001 |
| | ACVRL1 | 0.297 | 0.003 |

are more significantly associated with tumor purity compared to those in the SmCCNet network. Furthermore, a total of 139 molecular features are shared between the SGTCCA-Net and SmCCNet methods, with each method having 161 unique molecular features. Of the 161 unique molecular features in each network, 125 and 66 are significantly associated with tumor purity (FDR < 0.05) for SGTCCA-Net and SmCCNet, respectively.

We calculate the PageRank score for the SGTCCA-Net subnetwork and present the top 5 nodes for each molecular profile in Table 3. We observed that all of the molecular features in the table are significantly correlated with tumor purity. *ACVRL1* and *KAT2A* are the only two nodes with low correlation to tumor purity (FDR $\geq$ 0.001). To check whether these nodes contribute to the network, we examine the correlation between these two nodes and other molecular features of the subnetwork by looking at their top 5 connections (based on Pearson's correlation) to other molecular features. We observed that *ACVRL1* is moderately correlated with *PCNA* ($\rho_{between-omics}$ = 0.474), which is highly correlated with tumor purity ($\rho_{omics-pheno}$ = −0.564) compared to other network proteins (rank 1/175 among all proteins) and *KAT2A* ($\rho_{between-omics}$ = 0.476), which is highly correlated with tumor purity ($\rho_{omics-pheno}$ = −0.324) compared to other network proteins (rank 9/175 among all proteins).

The top 5 ontologies from the enrichment analysis for the SGTCCA-Net final subnetwork (Fig 4a) include lymphocyte activation (49/186 genes enriched), positive regulation of immune response (45/204 genes enriched), adaptive immune response (38/145 genes enriched), regulation of leukocyte activation (46/227 genes enriched), and Adaptive Immune System (37/178 genes enriched). Compared to the enrichment analysis results from SmCCNet, we observe that the ontology results from both networks fall into similar categories. However, when only looking at the immune response-related ontologies, a stronger enrichment analysis result is identified in the SGTCCA-Net network (Fig A in S1 Appendix). Additionally, using MCODE, we observed 6 modules of the protein-protein interaction network as shown in Fig 4b. For example, in the first network module (red), one of the proteins is *CD3D*, which has prognostic potential for breast cancer and is associated with lymphocyte infiltration and immune checkpoints [54].

When running correlation analysis between network PC1 (only PC among the first 3 PCs with significant association to tumor purity) and cell type estimates from CIBERSORT, we found that the top 3 correlated cell types are T cells (CD8) ($\rho$ = –0.781), Macrophages (M1)

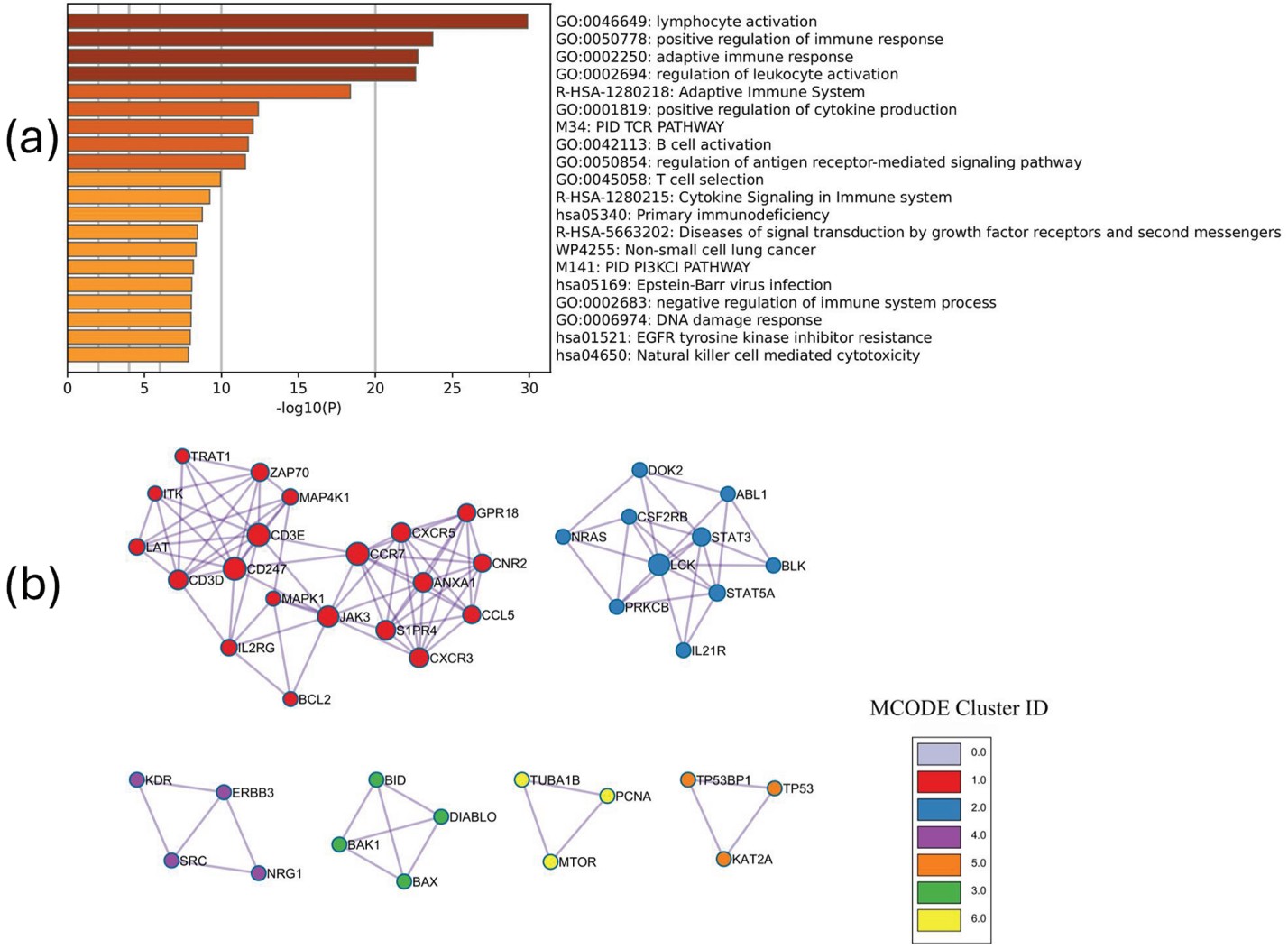

**Fig 4. Enrichment analysis results for TCGA breast cancer data with respect to tumor purity.** (a) The top pathways that are associated with the final network. (b) Protein-protein interaction (PPI) network for the multi-omics network from SGTCCA-Net with respect to tumor purity colored by clusters. Clusters are generated based on the Molecular Complex Detection (MCODE) algorithm (Cluster 0.0 has been hidden because of the cluster size).

(follicular helper) ($\rho = -0.528$), and T cells (follicular helper) ($\rho = -0.492$). The strong correlations with cell types involved in immune responses align with the top enriched ontologies displayed in Fig 4a, such as lymphocyte activation and adaptive immune response, indicating that the network is capturing key aspects of the tumor's immune landscape.

We ran MultiMiR on the miRNAs from each subnetwork to evaluate the quality and relevance of miRNA-gene connections. For the SGTCCA-Net network, among 1288 global network genes with non-zero canonical weight, 179/697 (13.9%/54.1%) are validated/predicted target genes of miRNAs in the subnetwork. These targets correspond to 202/1590 unique miRNA-gene connections, considering that each gene can be targeted by multiple miRNAs. Further quality assessment revealed that 75/1033 (37.1%/65.9%) validated/predicted miRNA-gene connections are supported by more than one database, indicating robust evidence for these interactions. In comparison, for the SmCCNet network, among 1024 global network

genes with non-zero canonical weight, only 102/212 (10.0%/20.7%) are validated/predicted target genes of miRNAs in the subnetwork. This results in 81/286 unique miRNA-gene connections, with only 35/134 (43.2%/46.9%) supported by more than one database. While SmC-CNet identifies fewer miRNA-gene connections, the lower proportion of validated or consensus connections suggests a more conservative feature selection approach, potentially prioritizing sparsity over coverage. This result indicates that SGTCCA-Net identifies a greater number of miRNA-gene connections, many of which share consensus across multiple databases, suggesting higher confidence and relevance. Enrichment analysis of the 179 validated targets in SGTCCA-Net subnetworks (Fig B in S1 Appendix) reveals similar ontology categories to the gene/protein enrichment analysis, including lymphocyte differentiation and the adaptive immune system. These findings highlight SGTCCA-Net's ability to uncover biologically meaningful and well-supported connections while balancing quality and coverage.

We examined the top combinations (4/3-way correlation) and found that the top 4-way correlation implies the top 3-way correlation in some cases (Table B in S1 Appendix). For example, it was observed that *hsa-mir-3133* and *KDR* dominate the 4-way relationship between gene, miRNA, protein, and tumor purity, which also implies the 3-way correlation between *hsa-mir-3133*, *KDR*, and tumor purity. Interestingly, *BEND4*, one of the genes present in the top 4-way correlation, is shown to be the predicted target of *hsa-mir-3133* with a predicted score of 0.728, and *KDR* is also the predicted target of *hsa-mir-3133* with a predicted score of 0.572. Through examining the 3-way correlation we can find that even though the top 3-way correlation could imply the 4-way correlation, the signal is generally weaker. For instance, it was found that *hsa-mir-142* dominates the gene-miRNA-tumor purity correlation, *PCNA* dominates both the gene-protein-tumor purity correlation and the miRNA-protein-tumor purity correlation, and *CCNB1* dominates the miRNA-protein-tumor purity correlation (details in Table B in S1 Appendix). However, it was observed that the highest 4-way correlation involved in *hsa-mir-142* is 2.405 (rank 94/631,680 across all 4-way correlation), in *PCNA* is 2.615 (rank 59/631,680 across all 4-way correlation), in *CCNB1* is 2.093 (rank 191/631,680 across all 4-way correlation). A particular example is the 3-way correlation that involves *CCNB1* and *hsa-mir-150*, which is observed in the top 10 3-way correlation between miRNAs, proteins and tumor purity, but the highest 4-way correlation involves these two molecular features is only 1.914 (rank 398/631,680). This suggests that some 3-way correlation combinations may be ignored by the 4-way correlation, and indicates the importance of incorporating the 3-way correlation into the model.

Lastly, the PageRank score for each molecular feature in the final subnetwork was calculated, and the top 10 molecular features from each molecular profile were selected for network visualization in Fig 5. It was noticed that higher connectivity occurs in proteins *PCNA* and *CCNB1* (both appear in Table 3 and Table B in S1 Appendix), and both of them are observed in the top 3-way miRNA-protein-tumor purity correlation and found as prognostic biomarkers for breast cancer [55,56].

We also tested the stability of the network by implementing a leave-one-out approach: During every iteration, we took out one subject and ran SGTCCA-Net on the remaining 104 subjects, generating 105 different multi-omics networks. After running SGTCCA-Net, we found a stable consistent core of the network: 79 molecular features appeared in every network, and 136 molecular features appeared at least 100 times out of 105 networks. All of these features are part of the network based on all subjects. Based on these results, we can conclude that the network's core structure is highly stable, demonstrating the reliability of SGTCCA-Net in identifying significant molecular features across different subsets of the data. Furthermore, of these 136 molecular features, only one is not significantly associated with tumor purity, while among the remaining 164 molecular features in the original network, 54 are

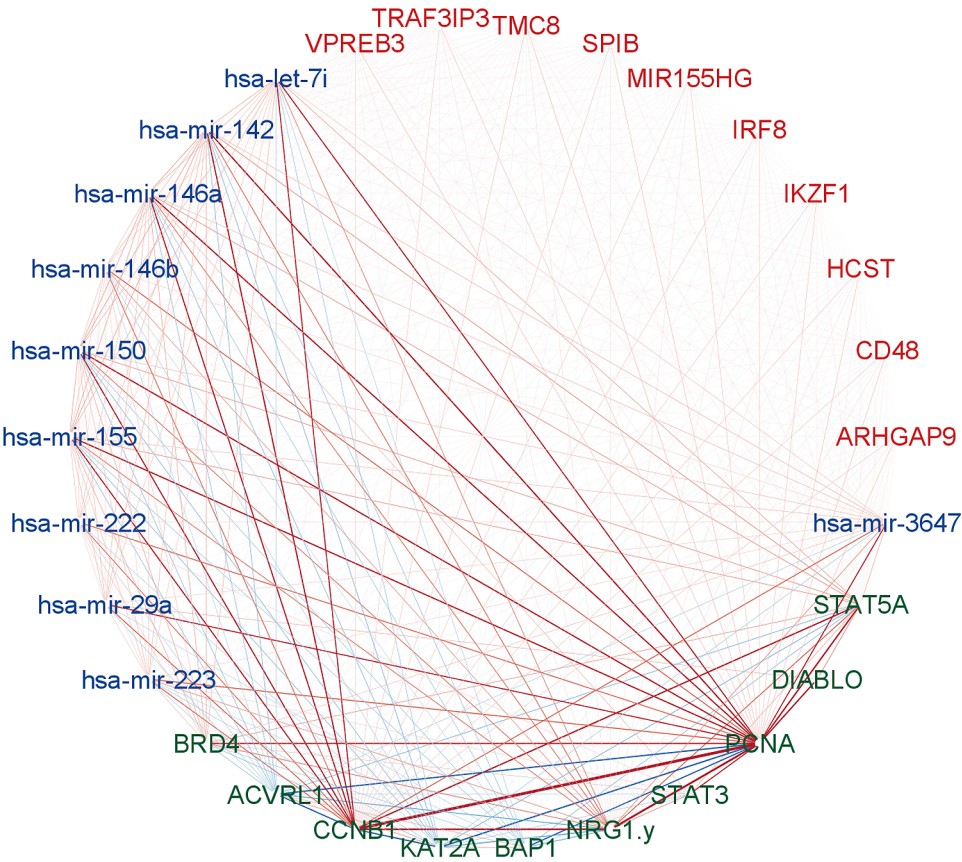

**Fig 5. SGTCCA-Net network with top 10 molecular features from each molecular profile.** Multi-omics network module for TCGA breast cancer data with respect to tumor purity. Nodes are genes (red), miRNAs (blue) and RPPAs (green). The edge color denotes positive correlation (red) or negative correlation (blue) between molecular features with the width denoting the strength of the connection. Edges are filtered based on Pearson correlation with a threshold of 0.2.

not significantly associated with tumor purity (FDR > 0.05). These results indicate that the strongest nodes obtained from SGTCCA-Net are less likely to be removed than the weaker nodes. These findings demonstrate the robustness of SGTCCA-Net in capturing key molecular features relevant to tumor purity, enhancing its potential utility in multi-omics research.

## 4.3. Relevant biological pathways found for COPDGene network analysis using SGTCCA-Net

After running the network analysis pipeline with SGTCCA-Net and SmCCNet for the $FEV_1$ phenotype, the final network size for SGTCCA-Net is 170, while SmCCNet has a final network size of 117. The NetSHy summarization score for SGTCCA-Net correlation with respect to FEV1 is 0.211, while for SmCCNet it is only 0.162. These are relatively large values as the maximum and 90th quantile of the univariate correlation between any of the molecular features and FEV1 are 0.28 and 0.12 respectively. In the SGTCCA-Net network, there are 58 molecular features with weak correlation with respect to $FEV_1$ (FDR > 0.05), while for SmCCNet, there are 82 such molecular features. To check whether each method includes these

nodes incorrectly or whether they contribute to the network, we extracted the NetSHy summarization score on these weak nodes only, extracted the first 3 PCs, and calculated the maximum correlation between PC score and $FEV_1$. For SGTCCA-Net, the maximum correlation is $-0.159$ ($p < 10^{-3}$), while for SmCCNet, the maximum correlation is only $0.090$ ($p = 0.053$). This implies that the weak nodes included in SGTCCA-Net are shown to be more related to the phenotype than the weak nodes included in SmCCNet. Furthermore, a total of 29 molecular features are shared between the SGTCCA-Net and SmCCNet methods, with each method having 141 and 88 distinct molecular features respectively. Of the 141 unique molecular features in the SGTCCA-Net network, 89 of them are significantly associated with $FEV_1$ (FDR < 0.05), while of 88 unique molecular features in the SmCCNet network, only 14 of them are significantly associated with $FEV_1$ (FDR < 0.05). In particular, *Troponin T* ($\rho = -0.278, p < 10^{-3}$) and *C-reactive protein* ($\rho = -0.190, p < 10^{-3}$) have a strong correlation with respect to $FEV_1$ and are not identified by SmCCNet, and studies have shown that elevated *Troponin T* levels during exacerbation are associated with death in patients with COPD [57,58], and more validation will need to be conducted to evaluate whether it is related to $FEV_1$ at the stable state.

We calculate the PageRank score for the SGTCCA-Net subnetwork and present the top 5 nodes for each molecular profile in Table 4. We observed that most of the molecular features in the table are significantly correlated with $FEV_1$ (FDR < 0.05), with exceptions such as *Neutrophil gelatinase-associated lipocalin* and *1-stearoyl-2-oleoyl-GPI (18:0/18:1)\**. To check whether these nodes contribute to the network, we examined the correlation between these two nodes and other molecular features of the subnetwork and observed that *Neutrophil gelatinase-associated lipocalin* is highly correlated with *N-acetylneuraminate* ($\rho_{between-omics}$ = 0.382), which is highly correlated to $FEV_1$ ($\rho_{omics-pheno}$ = $-0.134$) and *1-stearoyl-2-oleoyl-GPI (18:0/18:1)\** is highly correlated with s *1-(1-enyl-stearoyl)-GPE (P-18:0)\** ($\rho_{between-omics}$ = 0.509), which is highly correlated to $FEV_1$ ($\rho_{omics-pheno}$ = 0.151).

Pathways such as regulation of TLR by endogenous ligand (7/13 genes enriched) and neutrophil degranulation (23/102 genes enriched) are strongly associated with the severity of

**Table 4. Top 5 molecular features from each molecular profile for SGTCCA-Net based on the PageRank and their individual correlation with respect to** $FEV_1$ **for COPDGene data (with p-value).** Ensembl gene ID is included for genes and SomaScan sequence ID is included for proteins.

| Type | Name (ID) | Corr. | FDR |
|---|---|---|---|
| Gene (68) | HK3 (ENSG00000160883) | −0.177 | 0.004 |
| | MGAM2 (ENSG00000257743) | −0.123 | 0.019 |
| | S100A9 (ENSG00000163220) | −0.138 | 0.013 |
| | BCL6 (ENSG00000113916) | −0.121 | 0.029 |
| | IRAK3 (ENSG00000090376) | −0.134 | 0.013 |
| | Matrix metalloproteinase-9 (2579-17) | −0.149 | 0.012 |
| Protein (43) | calgranulin B (5339-49) | −0.122 | 0.029 |
| | Complement component C9 (14136-234) | −0.163 | 0.007 |
| | Kallistatin (3449-58) | 0.158 | 0.009 |
| | Neutrophil gelatinase-associated lipocalin (2836-68) | −0.095 | 0.060 |
| | 1-oleoylglycerol (18:1) | 0.114 | 0.028 |
| Metabolite (59) | 1-stearoyl-2-oleoyl-GPI (18:0/18:1)* | 0.094 | 0.062 |
| | 1-stearoyl-2-linoleoyl-GPI (18:0/18:2) | 0.143 | 0.013 |
| | 1-linoleoyl-GPA (18:2)* | 0.103 | 0.042 |
| | retinol (Vitamin A) | 0.093 | 0.064 |

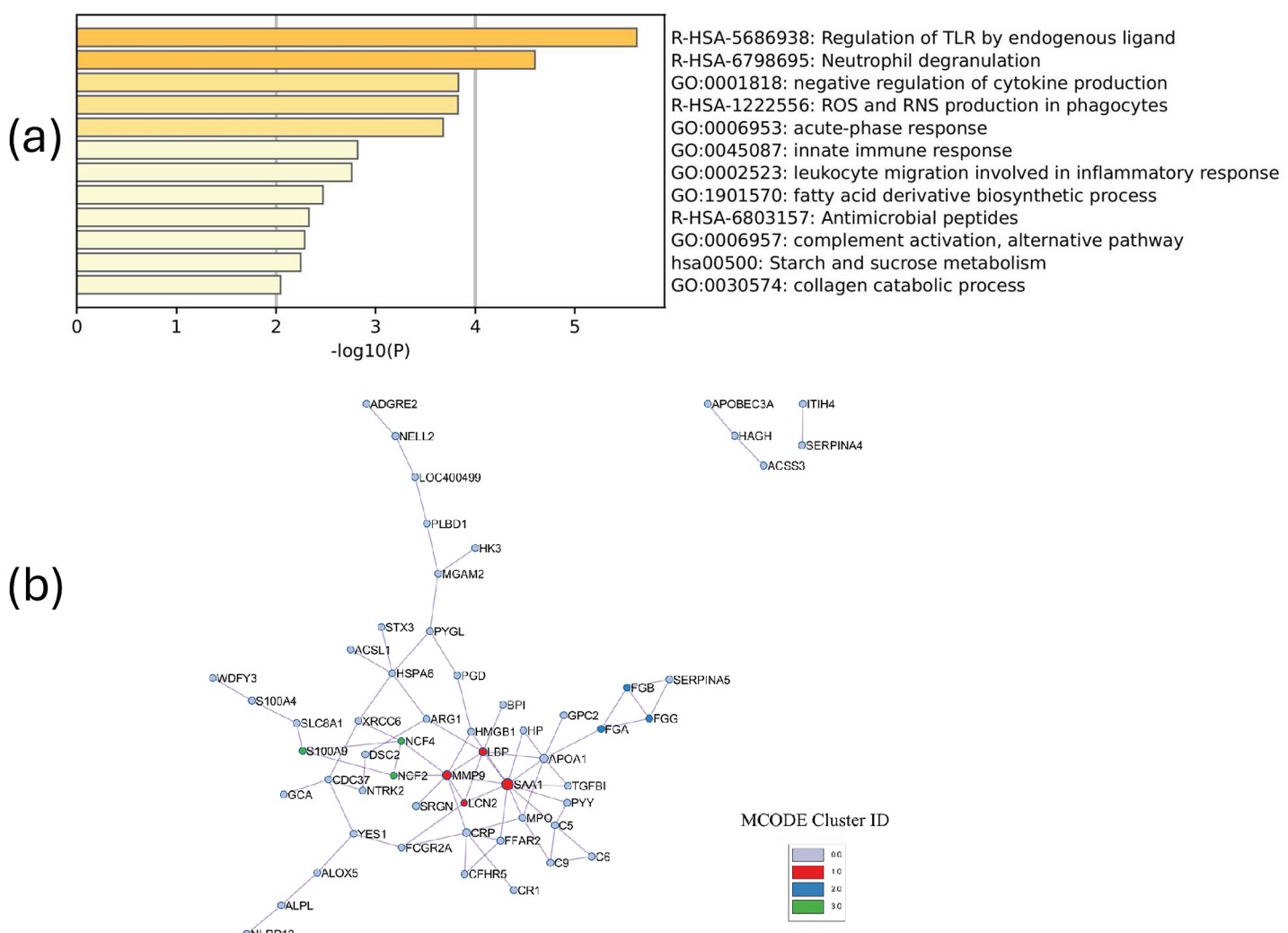

**Fig 6. Enrichment analysis results for COPDGene data with respect to** FEV$_1$. (a) The top pathways that are associated with the final network. (b) Protein-protein interaction (PPI) network for the multi-omics network from SGTCCA-Net with respect to FEV$_1$ colored by clusters. Clusters are generated based on the Molecular Complex Detection (MCODE) algorithm.

COPD from the enrichment analysis (Fig 6a) . Toll-like receptor (TLR) 2 is elevated in monocytes from individuals with COPD [59], and increased degranulation in COPD is found to increase on the surface of unstimulated neutrophils in patients with COPD and can cause additional airway damage in patients with COPD [60]. In particular, when checking the correlation between the network and cell type, we found that NetSHy PC1 is highly correlated with the percentage of neutrophil cells ($\rho$ = –0.767), which may explain the significance of the neutrophil degranulation pathway. However, we did not regress out cell type composition in advance as they can reflect disease state.

We ran the enrichment analysis on the metabolites of the SGTCCA-Net subnetwork and found that most of the top pathways are associated with phospholipid/glyerophospholipid and bile acid (Table D in S1 Appendix). Compared to non-smokers, current smokers have decreased phospholipid levels regardless of COPD status [61].

Fig 6b shows that there are three groups of protein-protein interaction based on MCODE, and the largest group is centered on the group with SAA1 and MMP9 (red) with the enrichment score of 1.5. SAA1 is related to the acute exacerbation of chronic obstructive pulmonary disease [62], and higher levels of MMP-9 correspond to a higher influx of neutrophils and lymphocytes, signaling an exacerbation of COPD in which a higher burden of MMP-9 is observed in the airways [63].

We examined the top combinations (4/3-way correlation) and found that the top 4-way correlation implies the top 3-way correlation in some cases (Table C1 in S1 Appendix.). For instance, it was observed that *MTCO1P12*, *Tyrosine-protein kinase Yes*, and *C-glycosyltryptophan* dominate the 4-way relationship between gene, protein, metabolite, and $FEV_1$, which also implies the 3-way correlation between *Tyrosine-protein kinase Yes* (the protein product of *YES1*), *C-glycosyltryptophan* and $FEV_1$. Through examining the 3-way correlation we can find that even though the top 3-way correlation could imply the 4-way correlation but the signal is generally weaker. For instance, it was found that *Matrix metalloproteinase-9* dominates the gene-protein-fev1 correlation, *1-stearoyl-2-oleoyl-GPI (18:0/18:1)\** dominates the gene-metabolite-fev1 correlation, and *C-glycosyltryptophan* and *iminodiacetate (IDA)* dominate the protein-metabolite-fev1 correlation (details in Table C in S1 Appendix.). However, it was observed that the highest 4-way correlation involved in *Matrix metalloproteinase-9* is 0.431 (rank 233/172,516 across all 4-way correlation), in *iminodiacetate (IDA)* is 0.580 (rank 30/172,516 across all 4-way correlation). A particular example is the 3-way correlation that involves *Matrix metalloproteinase-9* and other genes, we found that all top 10 3-way correlation between gene, protein, and $FEV_1$ involve this protein, but only 2 of the genes are shown in the top 1000 4-way correlation along with *Matrix metalloproteinase-9*. This suggests that some 3-way correlation combinations may be ignored by the 4-way correlation, and indicates the importance of incorporating the 3-way correlation into the model.

Furthermore, the PageRank score for each molecular feature in the final subnetwork was calculated, and the top 10 molecular features from each molecular profile were selected for visualization of the network in Fig 7. In particular, the C-reactive protein has been shown to be associated with poor lung function [64]; the expression level of S100A9 is elevated in patients with COPD, which triggers neutrophil degranulation and releases inflammatory and proteolytic enzymes [65].

We also repeated the analysis on $FEV_1$ percent predicted ($FEV_1PP$), which takes into account body habitus and race. After applying the SGTCCA-Net to $FEV_1PP$, we identified a final subnetwork with 43 features. In particular, 16 of these features exhibit a strong correlation with $FEV_1PP$ (with $|\rho| > 0.15$), and are all present in the $FEV_1$ final subnetwork. This significant overlap between the $FEV_1$ and $FEV_1PP$ subnetworks suggests a consistent molecular pattern that influences both metrics. Furthermore, Metascape enrichment analysis highlighted similar patterns in both $FEV_1$ and $FEV_1PP$ networks, with shared pathways including acute-phase response, inflammatory response, and neutrophil degranulation (Fig C in S1 Appendix). Interestingly, these molecular features generally show a higher correlation with $FEV_1PP$ than with $FEV_1$ (detailed in Table E in S1 Appendix.). This observation demonstrates that the associations between top-correlated molecular features and pulmonary function remain robust, unaffected by variations in body habitus and race.

## 5. Discussion

In this work, we introduce a new multi-omics network analysis pipeline, Sparse Generalized Tensor Canonical Correlation Analysis Network Inference (SGTCCA-Net), designed for

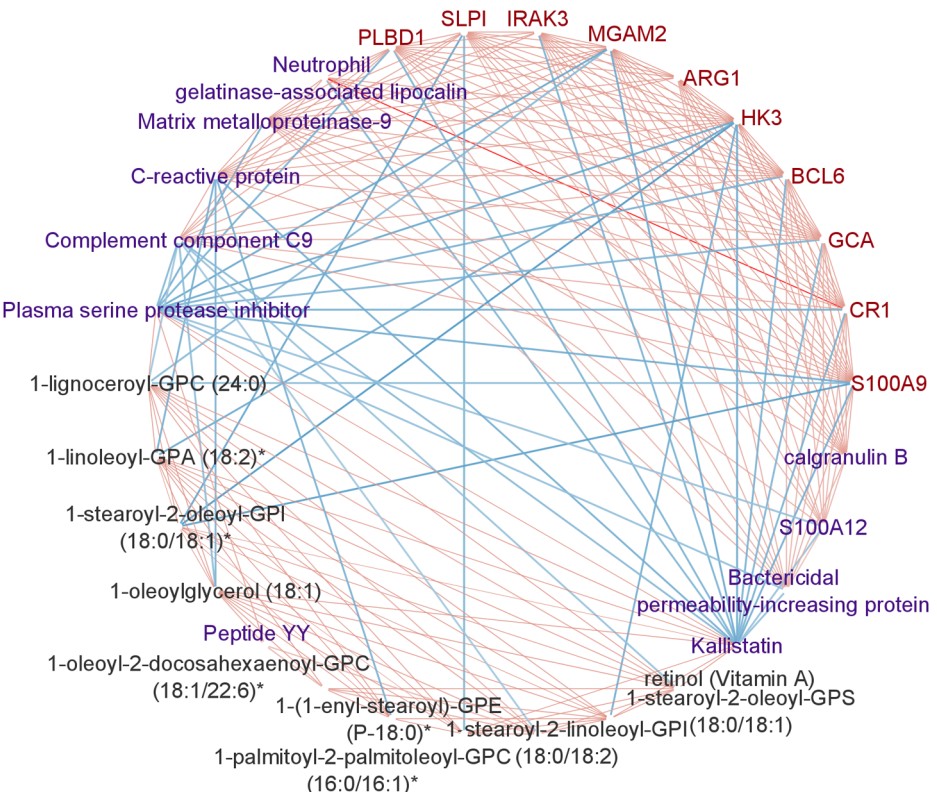

**Fig 7. SGTCCA-Net network with top 10 molecular features from each molecular profile.** Multi-omics network module for COPDGene data with respect to $FEV_1$. The red nodes stand for genes, the purple nodes stand for proteins, and the black nodes stand for metabolites. The width and color depth of the edge stands for the strength of the connection between two molecular features and the type of color stands for whether two nodes are positively correlated (red) or negatively correlated (blue). Edges are filtered based on Pearson correlation with a threshold of 0.2.

biomarker identification through both higher-order and lower-order correlations, thereby creating molecular interaction networks pertinent to specific phenotypes. We propose a new approach, Sparse Generalized Tensor Canonical Correlation Analysis (SGTCCA), which extends Tensor Canonical Correlation Analysis to accommodate multiple correlation structures at once, ensuring sparsity via the biased subsampling technique. Additionally, we bring forth a novel network analysis and network pruning algorithm to derive multi-omics subnetwork modules associated with the phenotype of interest.

Our proposed pipeline simplifies the process for users by requiring only the specification of correlation structures of interest, typically related to traits of complex diseases. Based on these defined correlation structures, the SGTCCA algorithm identifies molecular features that are involved in one or more of these relationships. Following this, the network analysis step reconstructs the interaction networks in a 2D space, enabling the visualization and interpretation of the internal biological mechanisms that link molecular features to the disease. This approach provides a clear and comprehensive understanding of the molecular interactions driving complex diseases, making the results more interpretable and actionable for researchers.

Our proposed method demonstrated its effectiveness by successfully capturing both higher-order and lower-order correlations within multi-omics data, leading to the construction of robust multi-omics networks. The results obtained from both simulation studies indicate that SGTCCA-Net effectively identifies phenotype-specific molecular interactions even in the presence of noisy data. The higher AUC scores observed across various scenarios confirm the reasonableness of the results, as SGTCCA-Net consistently outperformed other comparable methods SmCCNet and DIABLO. These findings validate the ability of the methodology to uncover complex molecular relationships. Additionally, we applied SGTCCA-Net to both COPDGene and TCGA breast cancer datasets, which demonstrates its robustness and practical utility in real-world scenarios. In the TCGA breast cancer data, SGTCCA-Net successfully identified phenotype-specific networks related to tumor purity, demonstrating superior performance compared to SmCCNet, with a higher NetSHy network summarization score and stronger correlations between the molecular features and tumor purity. These results indicate the method's ability to capture intricate multi-omics relationships that are critical for understanding cancer biology. Similarly, in the COPDGene dataset, SGTCCA-Net effectively uncovered networks associated with pulmonary function, revealing key molecular interactions that contribute to chronic obstructive pulmonary disease (COPD). Interestingly, for both studies, the biomarkers identified are not limited to those with strong phenotype correlations; we also identified molecular features with weak phenotype correlations that are highly connected to other nodes and contribute to the network. Furthermore, we validated our findings using various techniques, including enrichment analysis of biological pathways and miRNA-target relationships. These validations confirm the enhanced performance of SGTCCA-Net over SmCCNet and the ability to find relevant pathways. Furthermore, we also demonstrate the top higher-order correlation associated with the final subnetwork. These applications demonstrate that SGTCCA-Net not only produces reasonable and reliable results but also offers significant insights into complex diseases, making it a valuable tool for multi-omics network inference in diverse biomedical research contexts.

One potential drawback of higher-order correlation is that it measures the simultaneous relationship among multiple features, making it difficult to determine the directionality of effects through mathematical formulation alone. However, a post-hoc analysis can be conducted on the higher-order correlation results to examine the direction of effect for each pair of features. For example, after identifying the 4-way correlation between a gene, a protein, a metabolite, and a phenotype, a post-hoc analysis can be performed to look at the pairwise Pearson's correlation between each pair of features, such as the gene and protein. This approach allows researchers to investigate the directionality of relationships between individual pairs of features, providing a clearer understanding of the underlying biological interactions.

Beyond its primary use in multi-omics network inference, SGTCCA has broader applications and can be useful for multi-view dimensional reduction tasks. It can be used to project omics data into a shared lower-dimensional space, suitable for both supervised tasks, such as disease classification, and unsupervised tasks, such as subtyping.

## Supporting information

**Fig A in S1 Appendix. Enrichment analysis ontology results based on the joint set of gene/protein for TCGA breast cancer data and COPDGene data for SmCCNet subnetwork.** The top pathways that are associated with the final network of SmCCNet based on

Metascape. a: Top ontology pathways for TCGA breast cancer data with respect to tumor purity; b: Top ontology pathways for COPDGene data with respect to $FEV_1$.
(PDF)

**Fig B in S1 Appendix. Results of the enrichment analysis ontology based on the validated miRNA target genes for the TCGA breast cancer data for the SGTCCA-Net subnetwork.** Top ontology pathways based on validated miRNA target genes for the breast cancer data from TCGA with respect to tumor purity.
(PDF)

**Fig C in S1 Appendix. Enrichment Analysis Results of SGTCCA-Net for $FEV_1$ Percent Predicted:** This figure focused on pathways enriched in the joint gene/protein set derived from the COPDGene data, specifically within the SGTCCA-Net subnetwork associated with $FEV_1$ Percent Predicted. The identified pathways provide insights into the molecular mechanisms linked to pulmonary function as measured by $FEV_1$ Percent Predicted.
(PDF)

**Text A in S1 Appendix. Higher-order Covariance Tensor for both Odd and Even Number of Views.**
(PDF)

**Text B in S1 Appendix. Proof of Theorem 1.**
(PDF)

**Text C in S1 Appendix. Sparse Generalized Tensor Canonical Correlation Analysis (SGTCCA) Algorithm.**
(PDF)

**Text D in S1 Appendix. Biased Subsampling Covariance Density.**
(PDF)

**Text E in S1 Appendix. Network Analysis and Network Pruning Algorithm.**
(PDF)

**Text F in S1 Appendix. Additional Detail of Simulation Study Setup.**
(PDF)

**Table A in S1 Appendix. Simulation Setup Table.** Summary table of all simulation scenarios. Starting with 3 omics data simulation settings: (1) normal latent factors, (2) highly right-skewed latent factors, and (3) noisy omics data. Each of these settings has 3 cases: case 1 with all phenotype-specific correlation structures; case 2 with only 4-way phenotype-specific correlation structure; and (3) case 3 with 3-way and pairwise phenotype-specific correlation structure. All these cases will be evaluated based on sample sizes of 100 and 200. $\Sigma$ represents a diagonal matrix with diagonal values to be 1.
(PDF)

**Table B in S1 Appendix. Top Network Higher-order Correlation for TCGA Breast Cancer Network.** Top 5 network higher-order correlation of breast cancer data of different correlation structures.
(PDF)

**Table C in S1 Appendix. Top network higher-order correlation for the COPDGene network.** Top 5 network higher-order correlation of COPDGene data of different correlation structures.
(PDF)

**Table D in S1 Appendix. Top Metabolite Enrichment Pathways.** Top 10 pathways of metabolite enrichment analysis result from IMPaLa for SGTCCA-Net subnetwork. (PDF)

**Table E in S1 Appendix. Overlap Molecular Features between** $FEV_1$ **and** $FEV_1$ **Percent Predicted Network** Molecular features overlap between $FEV_1$ and $FEV_1$ percent predicted network generated from SGTCCA-Net (where absolute correlation with phenotype >0.15). (PDF)

**Table F in S1 Appendix. Summary of Mathematical Notations and Symbols.** (PDF)

## Acknowledgments

We would like to sincerely thank all the authors for their valuable contributions to this article. Their expertise and input have been instrumental in bringing this work to fruition. Additionally, we would like to acknowledge BioRender (https://www.biorender.com) for providing the platform used to create Fig 3, which visually supports our findings.

## Author contributions

**Conceptualization:** Weixuan Liu.

**Data curation:** Weixuan Liu.

**Formal analysis:** Weixuan Liu.

**Funding acquisition:** Russell P. Bowler, Farnoush Banaei-Kashani, Katerina J. Kechris.

**Investigation:** Katerina J. Kechris.

**Methodology:** Weixuan Liu, Katerina J. Kechris.

**Project administration:** Katerina J. Kechris.

**Resources:** Katerina J. Kechris.

**Software:** Weixuan Liu.

**Supervision:** Katerina J. Kechris.

**Validation:** Weixuan Liu.

**Visualization:** Weixuan Liu.

**Writing – original draft:** Weixuan Liu, Katerina J. Kechris.

**Writing – review & editing:** Weixuan Liu, Katherine A. Pratte, Peter J. Castaldi, Craig Hersh, Russell P. Bowler, Farnoush Banaei-Kashani, Katerina J. Kechris.

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
