## [Decision Letter · Decision Letter 0]

12 Jun 2024

Dear Mr. Liu,

Thank you very much for submitting your manuscript "A Generalized Higher-order Correlation Analysis Framework for Multi-Omics Network Inference" for consideration at PLOS Computational Biology.

As with all papers reviewed by the journal, your manuscript was reviewed by members of the editorial board and by several independent reviewers. In light of the reviews (below this email), we would like to invite the resubmission of a significantly-revised version that takes into account the reviewers' comments.

We cannot make any decision about publication until we have seen the revised manuscript and your response to the reviewers' comments. Your revised manuscript is also likely to be sent to reviewers for further evaluation.

Sincerely,

Jie Li

Academic Editor

PLOS Computational Biology

Mark Alber

Section Editor

PLOS Computational Biology

Reviewer's Responses to Questions

**Comments to the Authors:**

Reviewer #1: In “A Generalized Higher-order Correlation Analysis Framework for Multi-Omics Network Inference”, the authors present a new method for constructing small networks tied to defined phenotypes (or other descriptive data) that represent multi-view high dimensional data.

It’s an important avenue for research; multi-view data integration is extremely difficult, in many ways, but also rapidly becoming more common and necessary for analysis. Tools will therefore be very welcome.

The authors provide two worked examples, one being TCGA breast cancer data and the other being a COPD data set. In both cases, the algorithm produces fairly small networks, composed of connected node-features representing the multi-view, and show the networks are more closely related to the selected phenotypes compared to a competitive algorithm.

Thoughts:

(A) Multi-view data integration is especially difficult in biomedical situations where the data depends on tissue biopsies (or other tissue samples). The wet bench assays that produce the data, by nature destroy the samples, which make true multi-view data nearly impossible. There are some exceptions with more recent assay types such as CITE-sea and image based spatial sequencing data, but they are not completely mature data types.

In many cases, tissue structure related to disease tends to be high heterogeneous in its structure — that can relate to disease progression or subtype— and by splitting samples or using unrelated samples as multi-views, there will naturally be some disagreement or noise, discord, between the views because they are not views of the same bulk collection of cells. I expect this application is more difficult compared to other applications where there really is different views of exactly the same thing (like scans of the sun).

Then what I wonder about is the stability of the result. Is the resulting network robust? I think this could be explored by performing a leave-one-out procedure, where you drop one sample, run the pipeline, and compare the collection of resulting networks. Is there a large stable core of connected nodes? Do the weak nodes fall off more often than strong nodes? Maybe not. It would be surprising, but neat, if weak nodes turned out to be sample with variability in the sample set.

(B) In equation 1. It would be good to start out slow and bring the readers up to speed. So, right away defining things like w and X. What are their dimensions? What is the dimension of the output? Also you have parameters on the left hand side (w_1, w_2) and it seems like it should be more like F(w_1, w_2)=… ? Same for equation 3, there are parameters on the left hand side. Generally, just read through all the written equations one more time, checking for little details.

(C) At line 80, you bring up the fact that higher order correlations may not be composed of multiple pair wise, or lower order, correlations. This could be expanded on with a concrete example, because this comes up later, and by explaining it here, the first time the reader sees it, then when it’s encountered later on, the reader already has a good grasp on this. It may not be intuitive. Plus, it goes in the other direction also, lower order correlations are not necessarily always contained in the higher order correlations. This is important as it comes up in the simulation results. The real difference comes from using high and low order correlations in parallel.

(D) In section 0.6.1, the difficulty in interpretation of directionality of higher order correlations is briefly discussed. It’s an unfortunate fact, because from a modeling standpoint, that directionality is what we want. When gene x goes up, what happens to protein y and metabolite z?

So then, is the problem addressed by making the correlations absolute value (eq 4.)? Maybe I don’t see where this is directly addressed.

(E) At the end of eq 12., after the s.t., should that “i=g,pr,m,ph” be “j=…”? Because there’s so “i” in the formula. In eq 13., it appears to be a “j”.

(F) Line 459, to me a correlation of 0.5 does not seem like it can be called “highly” correlated. Maybe moderately correlated. Maybe compare it to the distribution of correlation values within network and outside the network? Does that make it “high”?

(G) It would be interesting to compare the features to cell type estimates from CIBERSORT (for example), which are already available for TCGA data. Or looking in single cell data for where certain genes are expressed, what cell types. Since the selected phenotype was tumor purity, the network is almost surely related to the quantity and type of cells other than tumor cells.

(H) Maybe table 4 could have an additional column, the proteins certainly need at least an ID or maybe a symbol.

(I) With regard to figure 6, have a look at “biofabrics” as a way of visualizing network structures. Might find it useful in cases like this.

(J) Figure 5b could use a legend to show how many and what colors are used for the cluster labels. It’s hard to see them as it is.

(K) In Appendix S1, equation (1) provides an equation for covariance. In the second paragraph you write "Eq 1 calculates the higher-order correlation between vectors", but that's covariance, correct? Perhaps review the language around this across the manuscript? There seems to be some exchangeability between the terms that should be corrected. When deriving the correlation, I would expect to see cov(U,V) / sqrt( var(U) var(V) ) someplace, maybe I'm missing it.

Minor:

⁃ Line 36, reference for SmCCA?

⁃ Unfortunate the algorithm is TCCA with TCGA data! Ha ha.

⁃ Line 152 “as follows:”

Reviewer #2: The manuscript by Liu et al. presents a novel framework for multi-omics network inference that addresses several limitations of existing methods. The proposed Sparse Generalized Tensor Canonical Correlation Analysis Network Inference (SGTCCA-Net) is an advancement of the previously developed Tensor Canonical Correlation Analysis (TCCA), incorporating higher-order correlations and several other improvements. SGTCCA-Net shows promise in effectively integrating multi-omics data, accounting for both higher-order and lower-order correlations, and improving computational efficiency. The manuscript provides an important contribution, especially considering the rapid accumulation of multi-omics data. However, several points need to be addressed for clarity:

1) It would be beneficial to include a figure illustrating the types of data SGTCCA-Net can handle, highlighting the dimensionalities of the data (e.g., genes vs. conditions/patients).

2) The description for cases 1-3 in Experiments 0.11, as explained in the caption of Table 1, is crucial. It demonstrates the superiority of SGTCCA-Net in handling three types of data. This explanation should be included in the main text and presented in an orderly fashion. Additionally, the corresponding case names should be labeled in Table 1.

3) In Experiments 0.11, the evaluation metric, Area Under the Receiver Operating Characteristic Curve (AUC-ROC), is not well defined in the manuscript or the supplementary information. A clear definition and ROC plots should be included. The authors should also explain how this metric evaluates the performance of the methods.

4) In Experiments 0.14, when using external software (e.g., MultiMiR), a brief description should be included to provide context.

5) The subheadings in the Results section currently refer to the dataset used, which is not particularly informative. The authors should consider using short sentences that describe the key take-home message of each section.

6) The numbering of the subheadings needs to be revised. Remove the “0.” prefix and simplify the numbering system for clarity.

Reviewer #3: The manuscript provides an insightful contribution to the field of multi-view dimension reduction. Below are some suggestions to enhance the clarity of the work:

The manuscript would benefit from explaining why other dimension reduction methods were not listed in Introduction 02. Providing a rationale for the selection of the chosen methods, and discussing why others were not considered, would enhance the reader's understanding. Highlighting the unique or suitable aspects of these methods would be particularly beneficial.

Lines 26-30: When mentioning more general solutions, it would be helpful to clarify whether these solutions are specific to Canonical Correlation Analysis (CCA) or applicable to other methods as well.

Describing the symbols used in the formulas throughout the manuscript would aid reader comprehension. Additionally, ensure that GTCCA and other abbreviations are clearly defined in both the figure and the main text. Clear explanations of each part of the figures will help readers easily understand their meaning.

Lines 35-36: Providing a more detailed explanation of SmCCA would help readers understand this concept better.

Explaining the principles behind the algorithms/proposed method would help readers appreciate the rationale for using this proposed method. Including diagrams to illustrate the process of algorithms in the methods section could enhance understanding and accessibility.

Line 134: Clarifying what is meant by the directionality (positive/negative) of higher-order correlation and why it is not interpretable would provide valuable insight.

Ensuring consistency in the row sizes of the data shown in A1 and A1 in S3 text.

Providing citations to support Theorem 1, if it has been previously validated, would ensure it has a strong mathematical foundation.

Clarifying why DIABLO was specifically chosen for comparison and highlighting its relevance and significance would improve the comparison discussion. Additionally, discussing other methods that could be used for comparison alongside SmCCNet and DIABLO would provide a more comprehensive evaluation of the proposed approach.

Including a thorough discussion of how the proposed method leads to the results and an analysis of whether the results from the data are reasonable. Discussing the methodology's effectiveness and the reasonableness of the results obtained would provide valuable context and insight.

**Have the authors made all data and (if applicable) computational code underlying the findings in their manuscript fully available?**

Reviewer #1: Yes

Reviewer #2: Yes

Reviewer #3: None

PLOS authors have the option to publish the peer review history of their article (what does this mean?). If published, this will include your full peer review and any attached files.

Reviewer #1: No

Reviewer #2: No

Reviewer #3: No
---

## [Decision Letter · Decision Letter 1]

4 Nov 2024

PCOMPBIOL-D-24-00109R1A Generalized Higher-order Correlation Analysis Framework for Multi-Omics Network InferencePLOS Computational Biology Dear Dr.  Liu, Thank you for submitting your manuscript to PLOS Computational Biology. After careful consideration, we feel that it has merit but does not fully meet PLOS Computational Biology's publication criteria as it currently stands. Therefore, we invite you to submit a revised version of the manuscript that addresses the points raised during the review process. Please submit your revised manuscript within 30 days. If you will need more time than this to complete your revisions, please reply to this message or contact the journal office at ploscompbiol@plos.org. Please include the following items when submitting your revised manuscript:* A rebuttal letter that responds to each point raised by the editor and reviewer(s). You should upload this letter as a separate file labeled 'Response to Reviewers'. This file does not need to include responses to formatting updates and technical items listed in the 'Journal Requirements' section below.* A marked-up copy of your manuscript that highlights changes made to the original version. You should upload this as a separate file labeled 'Revised Manuscript with Track Changes'.* An unmarked version of your revised paper without tracked changes. You should upload this as a separate file labeled 'Manuscript'. If you would like to make changes to your financial disclosure, competing interests statement, or data availability statement, please make these updates within the submission form at the time of resubmission. Guidelines for resubmitting your figure files are available below the reviewer comments at the end of this letter. We look forward to receiving your revised manuscript. Kind regards, Jie LiAcademic EditorPLOS Computational Biology Christoph KaletaSection EditorPLOS Computational Biology 

Feilim Mac Gabhann

Editor-in-Chief

PLOS Computational Biology

Jason Papin

Editor-in-Chief

PLOS Computational Biology

 **Journal Requirements:** **Additional Editor Comments (if provided):** PCOMPBIOL-D-24-00109

" A Generalized Higher-order Correlation Analysis Framework for Multi-Omics Network Inference" Revision 1 "

**Reviewers' comments:** Reviewer's Responses to Questions

**Comments to the Authors:**

Reviewer #1: All concerns have been addressed. Thank you! Nice work.

Reviewer #2: The authors have addressed all the questions that I had.

Reviewer #3: This work introduces a novel technique for multi-omics network analysis, building upon TCCA. The proposed method incorporates both higher-order and lower-order correlations, enhancing computational efficiency and applying sparsity to focus on the most relevant data. The technique is evaluated using TCGA Breast Cancer and COPD Gene datasets, demonstrating superior performance compared to existing algorithms. However, several points need clarification for better understanding.

1. line 176-177 You mention that the original TCCA faces challenges in interpreting the directionality of higher-order correlations. However, the proposed SGTCCA-Net seems to share a similar limitation regarding the interpretability of directionality. Could you explain what has been improved in SGTCCA-Net compared to the original TCCA regarding this limitation?

2. In Eq(5), show how covariance is calculated for an odd number of views in the main text would help highlight what sets your method apart from the original TCCA.

3. Each step in Figure 3 needs a bit more explanation. It would be easier to understand if the figure’s process is clearly described. Also, linking this figure with Algorithms 1 and 2 would help readers follow along better.

4. line 195, z_1,2,3=z1z2z3, Could you clarify why they are being grouped, what is the relationship between these variables?

5. There are several numerical expressions throughout the paper displayed as fractions (/) or as pairs (/,*) without sufficient explanation. If these numbers are not actual fractions, please consider rewriting them in a more appropriate format. Additionally, clarify whether these correspond to the definition of the minimally/maximally sized final subnetwork. For instance, in line 661, the meaning of (18:0/18:1) needs further elaboration to avoid confusion.

6. The comparison with SmCCNet mentions that it selects fewer validated/predicted target features and that the final network size is smaller. While this could suggest that selecting more features is better, it’s important to note that in sparse problems, quality is often prioritized over quantity. Some methods intentionally select fewer, more relevant features.

7. It would help readers if you explain what would be considered "good" values or ranges for the test results you present. Providing context will make it easier to interpret the results.

**Have the authors made all data and (if applicable) computational code underlying the findings in their manuscript fully available?**

Reviewer #1: Yes

Reviewer #2: Yes

Reviewer #3: None

PLOS authors have the option to publish the peer review history of their article (what does this mean?). If published, this will include your full peer review and any attached files.

Reviewer #1: No

Reviewer #2: No

Reviewer #3: No

---

## [Editor Report · Decision Letter 2]

31 Jan 2025

Dear Mr. Liu,

We are pleased to inform you that your manuscript 'A Generalized Higher-order Correlation Analysis Framework for Multi-Omics Network Inference' has been provisionally accepted for publication in PLOS Computational Biology.

Best regards,

Jie Li

Academic Editor

PLOS Computational Biology

Christoph Kaleta

Section Editor

PLOS Computational Biology

---

## [Editor Report · Acceptance letter]

PCOMPBIOL-D-24-00109R2

A Generalized Higher-order Correlation Analysis Framework for Multi-Omics Network Inference

Dear Dr Liu,

I am pleased to inform you that your manuscript has been formally accepted for publication in PLOS Computational Biology. Your manuscript is now with our production department and you will be notified of the publication date in due course.

With kind regards,

Anita Estes
